

# Impacts of temperature and soil characteristics on methane production and oxidation in Arctic polygonal tundra

Jianqiu Zheng[1], Taniya RoyChowdhury[1,2], Ziming Yang[3,4], Baohua Gu[3], Stan D. Wullschleger[3,5], David E. Graham[1,5]

[1] Biosciences Division, Oak Ridge National Laboratory, Oak Ridge, Tennessee, 37831, USA
[2] Now at Department of Environmental Science & Technology, University of Maryland, College Park, Maryland, 20742, USA
[3] Environmental Sciences Division, Oak Ridge National Laboratory, Oak Ridge, Tennessee, 37831, USA
[4] Now at Department of Chemistry, Oakland University, Rochester, Michigan, 48309, USA
[5] Climate Change Science Institute, Oak Ridge National Laboratory, Oak Ridge, Tennessee, 37831, USA

*Correspondence to*: David E. Graham (grahamde@ornl.gov)

This manuscript has been authored by UT-Battelle, LLC under Contract No. DE-AC05-00OR22725 with the U.S.
Department of Energy. The United States Government retains and the publisher, by accepting the article for publication, acknowledges that the United States Government retains a non-exclusive, paid-up, irrevocable, world-wide license to publish or reproduce the published form of this manuscript, or allow others to do so, for United States Government purposes. The Department of Energy will provide public access to these results of federally sponsored research in accordance with the DOE Public Access Plan (*http://energy.gov/downloads/doe-public-access-plan*).

**Abstract**

Methane ($CH_4$) oxidation mitigates $CH_4$ emission from soils. However, it is still highly uncertain whether soils in high-latitude ecosystems will function as a net source or sink for this important greenhouse gas. We investigated $CH_4$ production and oxidation potential in permafrost-affected soils from degraded ice-wedge polygons with carbon-rich soils at the Barrow

Environmental Observatory, Utqiaġvik **(**Barrow) Alaska, USA. Frozen soil cores from flat and high-centered polygons were sectioned into active layers, transition zones, and permafrost, and incubated at -2, +4 and +8°C to determine potential $CH_4$ production and oxidation rates. Organic acids produced by fermentation fueled methanogenesis and competing iron reduction processes responsible for most anaerobic respiration. Significant $CH_4$ oxidation was observed from the suboxic transition zone and permafrost of flat-centered polygon soil, which also exhibited higher $CH_4$ production rates during the

incubations. Although $CH_4$ production showed higher temperature sensitivity than $CH_4$ oxidation, potential rates of $CH_4$ oxidation exceeded methanogenesis rates at each temperature. Assuming no diffusion limitation, our results suggest that $CH_4$



oxidation could offset $CH_4$ production and limit surface $CH_4$ emissions, in response to elevated temperature, and thus should be considered in model predictions of net $CH_4$ fluxes in Arctic polygonal tundra.

## 1 Introduction

Arctic ecosystems store vast amounts of organic carbon in active layer soils and permafrost (Hugelius et al., 2014; Shiklomanov et al., 2010). Warmer air temperatures are increasing soil temperatures, annual thaw depths, and length of the thaw season, exposing more of this carbon to microbial degradation and mineralization (Shiklomanov et al., 2010; Schuur et al., 2015; Schuur et al., 2013). The potential carbon loss due to these direct effects is estimated to be 92±17 Pg carbon over the coming century (Schuur et al., 2015). In addition, thawing of ground ice and ice-wedge degradation cause ground

subsidence and significant changes in soil water saturation (Liljedahl et al., 2016), which could alter the future fluxes of carbon dioxide ($CO_2$) and methane ($CH_4$) to the atmosphere (Schädel et al., 2016). Understanding the factors that control these fluxes is key to predicting the greenhouse gas feedback on a future warming climate.

Arctic tundra acts as a large net source of $CH_4$, with a current estimation of the source strength ranging widely from 8 to 29

Tg C yr$^{-1}$ (McGuire et al., 2012). The large uncertainty associated with this estimation is due to the spatial and temporal complexity of the Arctic ecosystem. The unique polygonal ground in Arctic coastal plain tundra creates natural gradients in hydrology, snow pack depth and density, and soil organic carbon storage that control $CH_4$ fluxes (Liljedahl et al., 2016; Lara et al., 2015). Thermal contraction processes create cracks in the tundra soil, which can fill with water that freezes to produce massive ground ice (French, 2007). This ice forms wedges that create the borders of three dominant polygon types, defined

by their surface relief and subsurface hydrology: low-centered polygons (LCPs), flat-centered polygons (FCPs), and high-centered polygons (HCPs) (MacKay, 2000). Poorly drained LCPs are characterized by wet centers bordered by raised, relatively dry rims and wet troughs. FCPs lack the rims of LCPs and are drier (Wainwright et al., 2015). When ice wedges erode and water drains from the polygons, troughs subside and rims disappear to form drier HCPs. Methane emissions from wet and inundated LCP sites were about one order of magnitude higher than emissions from drier FCP and HCP sites

(Vaughn et al., 2016; Sachs et al., 2010). Although a number of factors, including vegetation height and plant composition (von Fischer et al., 2010), soil inundation (Sturtevant et al., 2012), thaw depth (Sturtevant and Oechel, 2013; Grant et al., 2017), and season (Chang et al., 2014) were suggested as explanatory factors for $CH_4$ flux variations, the huge differences in $CH_4$ flux between polygon types could not be fully explained by variations in moisture or temperature (Sachs et al., 2010; Vaughn et al., 2016). High concentrations of dissolved $CH_4$ in the deeper active layer, contrasted with low surface emissions

from FCPs and HCPs, suggest the importance of methane oxidation in these landscape features might be underestimated (Vaughn et al., 2016).





Soil $CH_4$ fluxes result from the net effect of microbial $CH_4$ production and oxidation, coupled with transport processes. Soils at the Barrow Environmental Observatory in Utqiaġvik (Barrow), Alaska experience a wide range of arctic temperatures, from -20 to +4°C (Shiklomanov et al., 2010). Soil respiration and methanogenesis continue at low temperatures close to 0 °C, even after the soil surface freezes trapping gas under ice. Therefore, substantial annual $CH_4$ and $CO_2$ emissions from the

Alaskan Arctic occur during the spring thaw (Commane et al., 2017; Raz-Yaseef et al., 2017; Zona et al., 2016). However, it is unclear how rapid temperature change in Arctic soils affects the opposing processes of $CH_4$ production and oxidation due to their nonlinear response to temperature fluctuations (Treat et al., 2015).

Methane oxidation mitigates terrestrial $CH_4$ emission. Up to 90% of $CH_4$ produced in the soil is consumed in the upper dry

layers of soil by aerobic $CH_4$-oxidizing bacteria (methanotrophs) before reaching the atmosphere (Le Mer and Roger, 2001). High-affinity methanotrophs oxidize substantial amounts of atmospheric methane in high Arctic mineral soils (Lau et al., 2015). Methane oxidation rates are usually greatest in oxic, surficial soils, and methane oxidation is known to occur under oxygen-limiting conditions as well (Roslev and King, 1996). In the classical model of $CH_4$ oxidation profiles, there is usually a vertical gradient of decreasing $O_2$ concentration in the top cm of the soil column that is inversely correlated with an

increasing gradient of $CH_4$ through the suboxic/anoxic active layer. The relative abundance of methanotrophs generally correlates with $CH_4$ oxidation activity at the soil surface, and methanogens are relatively abundant in the deeper layer where oxygen is limiting (Lee et al., 2015). Methane oxidation potential in peat bogs is highest immediately above the water table, while most $CH_4$ is produced below the water table (Whalen and Reeburgh, 2000). In contrast, methanogenic and methanotrophic communities can overlap in the rhizosphere (Liebner et al., 2012; Knoblauch et al., 2015), where roots or

*Sphagnum* create an oxic/anoxic interface providing a substantial amount of oxygen for methanotrophs (Laanbroek, 2010; Parmentier et al., 2011) and organic substrates for methanogenesis. The rates of $CH_4$ oxidation are mainly governed by the abundance and composition of methanotrophic microbial communities and environmental factors including $CH_4$ and $O_2$ availability, soil air-filled porosity and soil-water content (Preuss et al., 2013). Previous studies of boreal lakes and wetlands showed that $CH_4$ production is more sensitive to temperature fluctuations than $CH_4$ oxidation, as $CH_4$ oxidation rates

respond more rapidly to $CH_4$ availability than temperature increase (Liikanen et al., 2002; Segers, 1998). However, most temperature sensitivity measurements for $CH_4$ processes have been performed at mesophilic temperatures that are higher than typical Arctic soil temperatures.

Various studies have identified correlated factors affecting $CH_4$ and $CO_2$ production in permafrost ecosystems upon thawing,

but the oxidation of $CH_4$ is not considered in most incubation studies (Treat et al., 2015). The coupled response of methanogenesis and $CH_4$ oxidation to increased temperature is poorly understood. Similarly, the role of fermenters that produce organic acid substrates for competing processes of methanogenesis and iron reduction is not well characterized. To improve estimates of $CH_4$ production from soils across the permafrost zone, additional research on rates of $CH_4$ oxidation is needed to scale up results from laboratory studies and to better constrain $CH_4$ budgets from permafrost region. In this study,





we use the natural geomorphic gradient of FCP and HCP soils to explore the soil organic carbon decomposition pathways and specifically test the following hypotheses based on temperate ecosystem $CH_4$ dynamics to explain $CH_4$ production in organic Arctic soils.

(1) $CH_4$ production is localized in the oxygen-depleted subsurface while $CH_4$ oxidation occurs in the surface layers of oxic soil.

(2) $CH_4$ production is more sensitive to temperature increase than $CH_4$ oxidation and will likely exceed $CH_4$ oxidation in wet areas in response to warming.

## 2 Materials and Methods

### 2.1 Site description and soil sampling

The study site is located at the Barrow Environmental Observatory (BEO), Utqiaġvik (Barrow), Alaska as part of the Intensive Study Site areas B (High-Centered Polygon, HCP) and C (Flat-Centered Polygon, FCP) of the Next Generation Ecosystem Experiments (NGEE) in the Arctic project. The centers of HCPs in area B were covered by lichens, moss and dry tundra graminoids, while centers of FCPs in area C hosted wet tundra graminoids, mosses and bare ground (Langford et al.,

2016). Intact frozen soil cores from the centers of a water-saturated FCP (N 71° 16.791', W 156° 35.990') and a well-drained HCP (N 71° 16.757', W 156° 36.288') were collected with a modified SIPRE auger containing a sterilized liner, driven by a hydraulic drill during a field campaign in April 2012 (Herndon et al., 2015a; Herndon et al., 2015b; Roy Chowdhury et al., 2015). All samples were kept frozen during core retrieval, storage and shipment to Oak Ridge National Laboratory (ORNL, Oak Ridge, TN). The frozen cores were stored at -20 $^o$C until processing. The thaw depth measured in September 2012 in the

HCP center was 40 cm, and thaw depths in the FCP varied from 41-47 cm.

The frozen soil cores were inspected and processed inside an anaerobic chamber (Coy Laboratories, MI. $H_2 \leq 2\%$ and $O_2 < 1$ ppm). The soil core collected from the FCP center showed evidence of discontinuous organic matter depositions below the active layer mineral horizon and above the permafrost. This transition zone from 40-50 cm between active layer and

permafrost was attributed to episodic thawing and cryoturbation (Schuur et al., 2008; Bockheim, 2007). No equivalent transition zone was identified in the HCP core. Both cores were sectioned into layers of 10-cm increments, and each section was inspected for evidences of roots, undecomposed organic matter, and soil Munsell color recorded to qualitatively infer redoximorphic features. Gravimetric water content, pH in a potassium chloride slurry ($pH_{KCl}$), Fe(II) concentration, total carbon and nitrogen were measured in each section as described previously (Roy Chowdhury et al., 2015), and pore water

gas concentrations were measured using the method described in Sect. 2.2.



Based on similarities in the above mentioned geochemical properties, several soil sections were combined to represent the active layer, transition zone and upper permafrost. The top 10 cm of the cores contained substantial ice or plant material; these sections were not used for soil incubation studies. The next two sections (comprising 10-30 cm depth) were combined to represent the active layer, for both FCP and HCP cores. The FCP section from 40-50 cm depth provided transition zone

samples. Finally, sections from 50-70 cm depth were combined to represent permafrost layer soils for both cores. Soils from each layer were homogenized inside the anaerobic chamber using sterile tools and equipment. The homogenized soil from each layer was sub-sampled for microcosm incubation studies described below.

**2.2 Soil pore water gas measurements**

Dissolved gas ($CO_2$ and $CH_4$) concentrations in soil pore water were determined at each 10-cm depth-interval from the FCP

and HCP cores. A 1:1 (w:v) soil slurry was prepared by mixing 10 g wet soil in 10 mL de-ionized and de-gassed water under anoxic conditions inside an anaerobic chamber. The samples were then placed in 15 mL crimp-sealed serum vials (Wheaton, NJ). Vials were inverted and shaken at 4 ℃ for 12 h to allow for exchange between the dissolved and soil gas phases. Then, ~5 mL of the aqueous phase was exchanged with Ar (99.9 % purity) using a Gastight syringe (Hamilton, NV), and samples were manually shaken vigorously for 5 min to allow for equilibration between the aqueous and gas phase. Subsequently, a

500-µL headspace sample was drawn and immediately analyzed with an SRI 8610C gas chromatograph using the method previously described (Roy Chowdhury et al., 2015). The detection limit for $CH_4$ was 1 ppm$_v$. Concentrations of $CH_4$ and $CO_2$ were corrected for dissolved gases based upon temperature and pH-dependent Henry's Law constants (Sander, 2015).

**2.3 Soil microcosm setup, methane oxidation potential assay and soil chemical analyses**

**2.3.1 Temperature sensitivity of $CO_2$ and $CH_4$ production from unamended microcosms**

To investigate the temperature response of $CH_4$ production and overall organic carbon mineralization rates (measured as $CO_2$ production), homogenized soils described in section 2.1 from the FCP or HCP active layer, transition zone (only in FCP), and permafrost were used in incubations (Table S1). Replicate (n=9) microcosms of each homogenized soil layer were constructed in sterile 70 mL serum bottles sealed with blue butyl stoppers and aluminum crimps. Soils from FCP and HCP core layers were incubated under anoxic or oxic conditions, determined by the concentrations of reduced Fe(II) measured in

each layer (Howeler and Bouldin, 1971). Microcosms from active layers of both FCP and HCP cores were prepared in oxic conditions with ambient laboratory air. Microcosms from the transition zone and permafrost of FCP and the permafrost layer of HCP were set up inside the anaerobic chamber. Microcosms were flushed with $N_2$ three times after sealing to remove residual $H_2$ and $O_2$ from the headspace. All microcosms contained ~10 g wet soil, and they were incubated at -2, +4 or +8 ℃ for approximately 90 days. It is important to note that at -2 ℃ soil water remained unfrozen in these samples due to freezing

point depression (Romanovsky and Osterkamp, 2000).



### 2.3.2 Methane oxidation potential assay

Soil samples were incubated in oxic conditions supplemented with ample $CH_4$ substrate to measure $CH_4$ oxidation potential (Roy Chowdhury et al., 2014). After 0, 10 or 20 days of microcosm incubation described in section 2.3.1, three replicates corresponding to each layer and temperature were opened to set up $CH_4$ methane oxidation assays and further analyses.

Replicate (n=9) samples (about 2 g) from each time point were slurried in a 1:1 (w:v) ratio with autoclaved de-ionized water in 26 mL serum bottles under oxic conditions. A 1% $CH_4$ headspace was introduced into each crimp-sealed bottle by replacing 0.23 mL headspace with 99.99% $CH_4$. Samples were incubated at the same incubation temperature as the microcosms from which they were harvested. To eliminate gas-liquid phase transfer limitation, samples from FCP were shaken and incubated at 4 $^o$C, and samples from HCP were shaken and incubated at 8 $^o$C. Headspace $CO_2$ and $CH_4$

concentrations in soil incubations and methane oxidation assays were measured at 2 to 15 days' time intervals using gas chromatography as described in section 2.2.

### 2.3.3 Extractable ion analysis

Aliquots of soil samples from microcosms opened after 10, 20 and 90 days of incubation were extracted with either 10 mM

$NH_4HCO_3$ (pH ~ 7.3) or 0.1 M KCl (pH~ 5.0) solution for determining soil organic acids and extractable Fe(II), respectively. Briefly, $NH_4HCO_3$ extractions (12 h) were centrifuged for 15 min at 6500 *g*, and the supernatants were further filtered through 0.2 μm membrane filters before analysis. Filtered sample were analyzed for low-molecular-weight organic acids using a Dionex ICS-5000$^+$ system (Thermo Fisher Scientific, MA) equipped with an IonPac AS11-HC column with a KOH mobile phase. Fe(II) concentrations from filtered KCl extract samples were diluted and quantified using the HACH Ferrous

method 8146 (1,10-phenanthroline). Absorbance was measured at 510 nm using a DU 800 spectrophotometer (Beckman Coulter, CA). Soil pH was determined using the slurry method by mixing a 1:5, w:v ratio of soil to 1M KCl.

### 2.4 Rate estimation, temperature sensitivity and statistical analyses

Concentrations of $CH_4$ and $CO_2$ accumulated over time in soil microcosms were fitted with hyperbolic, sigmoidal, or linear functions (Roy Chowdhury et al., 2015). Rates of $CO_2$ production were calculated using derivatives of the best curve-fitting

equations with parameters listed in Table S1. To estimate temperature sensitivity, gas production rates at -2, +4, and +8 $^o$C were fit to the exponential Arrhenius equation, and a quotient of rates ($Q_{10}$) is calculated as described previously (Roy Chowdhury et al., 2015). Methane oxidation rates were calculated from the loss of headspace $CH_4$, which were best fitted with simple linear regression. All rate calculations are reported on per gram soil dry mass basis.

Changes of soil physicochemical properties were evaluated with one-way ANOVA, Tukey's Honest Significant Difference (HSD) test. The effect of soil layers (active, transition zone and permafrost) and incubation temperature (-2, 4 and 8 $^o$C) were





examined with Tukey's HSD test. All curve fittings and statistical analyses are performed with R 3.4.0 (The R Foundation

for Statistical Computing) and validated with Prism (GraphPad Software, ver. 7.0a).

**2.5 Calculation of net CH₄ emission**

Representation of CH₄ oxidation is based on Michaelis-Menten kinetics with a linear dependence on the biomass of

methanotrophs in Eq. (1) (Xu et al., 2015), while the CH₄ production rate is calculated from measurements directly using Eq.

(2).

$$R_{oxi} = B_{methanotrophs} \cdot V_{max,oxi} \left[ \frac{c_{CH_4}}{c_{CH_4}+K_{m,CH_4}} \right] \left[ \frac{c_{O_2}}{c_{O_2}+K_{m,O_2}} \right] \tag{1}$$

$$R_{pro} = B_{methanogens} \cdot V_{measure,pro} \tag{2}$$

where $B_{methanotrophs}$ and $B_{methanogens}$ represent the estimated biomass of methanotrophs and methanogens respectively. $K_{m,CH_4}$

and $K_{m,O_2}$ are the half saturation coefficients (mM) with respect to CH₄ and O₂ concentrations, respectively. Values of $K_{m,CH_4}$

and $K_{m,O_2}$ are 0.005 and 0.02, with ranges of 0.0005-0.05, 0.002-0.2, respectively (Riley et al., 2011). The maximum CH₄

oxidation rate $V_{max, oxi}$ and CH₄ production rate $V_{measure, pro}$ were obtained from the incubations. By assuming $R_{oxi}=R_{pro}$ , the

ratio of $B_{methanogens}$ to $B_{methanotrophs}$ can be calculated at different temperatures. Initial CH₄ and O₂ concentrations were

calculated from soil porewater dissolved gas measurement and soil air-filled porosity estimations.

**3 Results**

**3.1 Soil attributes and pore water characteristics**

Soil cores from FCP and HCP center positions showed distinct vertical profiles of soil moisture expressed as gravimetric

water content (g g⁻¹ dry soil). The soil core from FCP was characterized by a wet surface within the top 10 cm below ground,

a much drier intermediate layer between 10 to 40 cm, and a bottom layer below 40 cm with significantly higher water

content (Fig. 1). A similar water distribution has been recorded by continuous field measurements of volumetric water

content at the nearby NGEE_BRW_C soil pit monitoring site (http://permafrostwatch.org). Soil moisture from the HCP core

gradually increased from the top to the bottom (Fig. 1), consistent with field measurements of the water level in nearby

HCPs (Liljedahl et al., 2016). Fe(II) concentration showed a strong positive correlation with gravimetric water content in

both FCP and HCP cores ($R^2$=0.81, and $R^2$=0.91, respectively). The soil pH in FCP increased steadily with soil depth

($R^2$=0.95), with an average of 4.7. In the soil core of HCP, soil pH varied by 1.5 pH unit, with an average of 5.4. Overall, soil

moisture, Fe(II) concentration and pH increased with depth in the centers of both polygon types.





Dissolved $CO_2$ in soil pore water showed a similar general trend in both FCP and HCP cores. The concentration of dissolved $CO_2$ was between 0.2 to 0.6 μmol $g^{-1}$ dry soil in the first 40 cm and 0.9 to 1.6 μmol $g^{-1}$ dry soil below 50 cm. The highest dissolved $CH_4$ concentration was found between 40 to 50 cm in soil pore water of FCP, approximately 4 times the $CH_4$ concentration measured from the top 10 cm (Fig. 1). Significant $CH_4$ accumulation in soil pore water was found below 50

cm of the HCP core, while no dissolved $CH_4$ was detected above 50 cm. The active layers of FCP and HCP cores were both oxic, with low Fe(II) concentrations and minimal dissolved $CH_4$ in soil pore water, while deeper layers were more reduced and suboxic with significantly higher Fe(II) concentrations and dissolved $CH_4$ (Fig. S1).

The total carbon content of the FCP permafrost (31%) was nearly 66% greater than the active layer content and five-fold

more than the transition zone (Table S1). The total carbon content of active and permafrost layers of HCP were 21% and 17%, respectively, and substantially lower than that of the FCP permafrost. Inorganic carbon quantified as $CO_2$ released upon acid treatment was less than 0.001% for all layers of the FCP and HCP cores.

### 3.2 $CH_4$ production and oxidation rates

### 3.2.1 $CH_4$ production

Thawed FCP and HCP soil samples were incubated in microcosms at fixed temperatures to assess methanogenesis rates. To emulate the natural redox gradient, FCP and HCP active layer samples were incubated in oxic conditions, while transition zone and permafrost samples were incubated in anoxic conditions, as described in Sect. 2.3.1. $CH_4$ production was only observed in microcosms from the transition zone and permafrost of FCP, which were incubated under anoxic conditions.

$CH_4$ production was detected within 5 days after the anoxic incubations were set up (Fig. 2). Cumulative $CH_4$ concentrations at all temperatures were best fitted with a linear model (Table S2). Samples from the transition zone yielded about 10 times more $CH_4$ than permafrost layer samples.

$CH_4$ concentrations measured from both HCP layers, and the active layer of FCP were all below the detection limit of 1

ppm$_v$ (Fig. S2). Methanogenesis was unlikely to occur in the active layers from FCP and HCP as these microcosms were incubated under oxic conditions, and our calculation suggested that $O_2$ was not completely consumed after 90 days incubation (data not shown).

### 3.2.2 Methane oxidation potential

Potential rates of aerobic $CH_4$ oxidation were measured as described in Sect. 2.3.2 using freshly thawed soils (0 day and 5 days) and soils that had been previously incubated (10 days and 20 days pre-incubation) in the microcosms described in Sect.



2.3.1. Potential $CH_4$ oxidation rates in HCP soils were 80-90% lower than rates from the equivalent FCP soil layers (Fig. 3). $CH_4$ oxidation potentials from transition zone and permafrost soils were more than twofold higher than the active layer in freshly thawed FCP samples (Fig. 3a). Similar to FCP soils, freshly thawed HCP permafrost showed slightly higher $CH_4$ oxidation potentials than active layer soil (Fig. 3c).

Pre-incubated soils from FCP microcosms showed similar $CH_4$ oxidation potentials to newly thawed soils at $+8\,^{o}C$ (Fig. 3). However, the potential increased significantly in transition zone samples incubated at $-2\,^{o}C$ ($p$=0.01), perhaps due to methanotroph biomass increasing in response to methanogenesis during the 20 day pre-incubation. HCP permafrost layer samples obtained after 10 days of anoxic incubation also show higher $CH_4$ oxidation potentials (*$p<0.05$*, Fig. 3c, 3d).

**3.3 Temperature sensitivity of $CH_4$ production and oxidation**

Potential $CH_4$ production rates from both transition zone and permafrost of FCP were significantly higher at $+8\,^{o}C$ than $-2\,^{o}C$ ($p < 0.01$, t-test). Transition zone soil showed a stronger temperature response, with a $Q_{10}$ value of $4.2 \pm 0.9$, compared to $1.7 \pm 0.7$ for permafrost.

The temperature dependency of $CH_4$ oxidation was consistent among soils layers. $CH_4$ oxidation potentials significantly increased when the incubation temperature increased from $-2$ to $+8\,^{o}C$ in active layer soil ($p < 0.01$), with a $Q_{10}$ value of $1.7 \pm 0.3$. Potential rates also increased with temperature for the transition zone and permafrost soils despite more variability: $Q_{10}$ values were $2.0 \pm 1.6$ and $1.7 \pm 0.5$, respectively. HCP samples had relatively low rates of $CH_4$ oxidation. Significant temperature response was observed in pre-incubated permafrost samples from HCP microcosms ($p$=0.01). This soil sample also demonstrated the greatest temperature sensitivity, with a $Q_{10}$ value of $6.1 \pm 1.7$.

**3.4 $CO_2$ production and its temperature sensitivity**

Soils in all of the microcosm incubations produced $CO_2$, by aerobic respiration under oxic conditions or by anaerobic respiration and fermentation under anoxic conditions. $CO_2$ production started immediately (day 1) in microcosms of FCP samples (Fig. 4). Soils from the active layer were incubated under initial oxic conditions, and these oxic soils produced 10 times more $CO_2$ after prolonged incubations compared to soils from transition zone and permafrost samples that were incubated under anoxic conditions. $CO_2$ accumulation was best modelled by a hyperbolic function, except the active layer soil incubated at $-2$ or $+4$ °C where the best fit was a linear function (Table S3, Fig. 4). This exception is likely due to continuous aerobic respiration at lower temperatures, indicating that substrate limitation was not reached within 90 days. The transition zone had the slowest $CO_2$ production rates and least carbon loss via $CO_2$, in contrast to its relatively high rate of methanogenesis.

30




CO$_2$ production in microcosms of HCP samples showed a different pattern from FCP samples. For active layer soils, the cumulative CO$_2$ production from aerobic incubations of HCP soils was an order of magnitude lower than FCP soils, despite similar total C content (Fig. 4). The same difference in cumulative CO$_2$ production was also observed from permafrost

incubated under anoxic conditions. The HCP cumulative CO$_2$ production profiles were best fitted with a sigmoidal model, compared to the hyperbolic model that best fit FCP data (Table S3). A prolonged delay in CO$_2$ accumulation was observed in both HCP active and permafrost layer samples. CO$_2$ production started about 10 days after the microcosm setup in the active layer and reached a maximum rate at 30 days (+4 and +8 °C) or 75 days (-2 °C). The delay was longer in permafrost incubations, with a rapid increase after 40 to 50 days followed by a plateau. Therefore, microorganisms mineralized more

carbon from FCP soils than HCP soils, and CO$_2$ production began sooner in FCP than HCP soil incubations.

Temperature showed significant effects on CO$_2$ production from the active layer and transition zone of FCP during 90 day incubations ($p<0.01$ for each layer) (Fig. 4). FCP soils incubated at +8 °C produced substantially more CO$_2$ than those incubated at lower temperatures. In the permafrost layer of FCP, CO$_2$ production was significantly higher at +8 °C compared

to 2 °C ($p<0.05$, ANOVA with Tukey's multiple comparisons test). The temperature sensitivity for CO$_2$ production in FCP soils was highest for oxic active layer FCP samples ($Q_{10} = 5.8 \pm 1.8$) and lower for anoxic transition zone and permafrost samples ($Q_{10} = 0.9 \pm 0.6$ and $2.2 \pm 1.0$, respectively estimated for day 1). The $Q_{10}$ values of HCP active and permafrost layers could not be readily estimated from sigmoidal models due to different lag periods, but were empirically estimated at $2.7 \pm 0.02$ and $2.6 \pm 1$, respectively. Therefore, the FCP active layer aerobic mineralization was most sensitive to a

temperature rise, while the anaerobic processes in the transition zone and permafrost had a temperature sensitivity typical of many soils (discussed below).

### 3.5 Organic acids production and iron reduction in FCP soils

Organic acids were produced as intermediate metabolites during microbial degradation of organic matter, and they probably

fueled methanogenesis and iron reduction. Specific organic acids were analyzed from soil extracts from FCP transition zone and permafrost (Table 1). The most abundant organic acids included formate, acetate, propionate, butyrate, and oxalate, consistent with previous analyses from LCP soils (Herndon et al., 2015a). Trace amount of lactate, pyruvate, and succinate were detected, with concentrations less than 0.05 μmol g$^{-1}$ soil. Concentrations of dominant organic acids measured in the permafrost were approximately 10 times higher than those measured in the transition zone. The difference was associated

with lower total carbon content (5.8%) in the transition zone than that in the permafrost (30.8%). Significant increases in acetate concentration after the incubation were observed from both transition zone ($p=0.02$) and permafrost ($p=0.02$) layers, while formate and propionate decreased by up to 40% in both layers.



Incubation temperature showed a profound impact on the acetate dynamics. In the transition zone, acetate concentration increased by 56%, 142%, and 156% at -2, 4, and 8°C, respectively after the incubation. Acetate concentrations in permafrost samples increased from high initial values by 17%, 39% and 53%, respectively.

To compare the changes in organic acids over time, total organic carbon contained in formate, acetate, propionate, butyrate and oxalate products was calculated ($T_{OA}$, μmol C $g^{-1}$), and the changes during the incubation were plotted for each layer of FCP (Fig. 5). $T_{OA}$ drastically dropped to near zero in the active layer due to the aerobic decomposition. Under anaerobic conditions, $T_{OA}$ generally increased during the incubations, except for permafrost incubated at 4 °C.

Significant changes in Fe(II) concentrations were observed from anoxic incubations of FCP samples over the incubation period (Fig. 6). In the transition zone, Fe(II) concentrations stayed at initial levels during the first 20 days of anoxic incubation, and then increased significantly from ~50 μmol $g^{-1}$ to ~100 μmol $g^{-1}$ during the 20 to 90 days incubation period at -2, 4 and 8 °C. In the permafrost, the highest level of iron reduction within the first 20 days was observed in samples incubated at -2 °C. Between 20 and 90 days, the highest iron reduction rates were observed in samples incubated at 8 °C.
The estimated $Q_{10}$ values were 1.2 and 1.3 for transition zone and permafrost, respectively. If we assume that iron reduction is coupled to acetate oxidation to produce $CO_2$, stoichiometric calculations suggest that iron reduction could account for 96% and 70% of $CO_2$ produced in the transition zone and permafrost at -2 °C. At 8 °C iron reduction could account for 74% and 61% of acetate oxidation in the transition zone and permafrost, respectively.

**4 Discussion**

The widespread ice-wedge degradation in the Arctic causes morphological succession and hydrological changes in tundra ecosystems (Liljedahl et al., 2016). FCP and HCP features represent successively more degraded polygons. Despite clear geomorphological differences between polygon types that affect drainage, vegetation and snow cover, the FCP and HCP active layers were similar in terms of their gravimetric water contents, pH, SOC, and Fe(II) concentrations. Permafrost from
both polygons contains more water, dissolved $CH_4$ and Fe(II) than active layer soils indicating more reducing environments. Concentrations of $CH_4$ measured in soil pore water from FCP and HCP cores increased with depth. These results are consistent with field measurements of dissolved $CH_4$ in soil pore water sampled using piezometers during the thaw season (Herndon et al., 2015b). The disconnect between $CH_4$ trapped in the permafrost layer and trace amounts of dissolved $CH_4$ in active layer samples suggests $CH_4$ oxidation in the upper active layer. Therefore, we began this work with the following
hypotheses for $CH_4$ production and oxidation, which were informed by previous studies of methane cycling in temperate ecosystems but untested in the Arctic. (1) $CH_4$ is produced in the more reduced subsurface, and consumed by methane





oxidizers at the upper section of the soil column where $O_2$ is available. (2) Methane production has higher temperature sensitivity than $CH_4$ oxidation, and is likely to exceed the $CH_4$ consumption rate in warmer temperature.

Aerobic $CH_4$ oxidation is usually assumed to be limited by $O_2$ diffusion. Therefore, near-surface layers of soil and the
rhizosphere would be expected to have the highest $CH_4$ oxidation activity (Shukla et al., 2013; Gulledge et al., 1997). The relative abundance of the *pmoA* marker gene for $CH_4$ oxidation decreased with soil depth in HCP trough soils (Yang et al., 2017) and in permafrost-affected soils from the Canadian Arctic (Frank-Fahle et al., 2014), consistent with that conceptual model. Thus, we would expect that the active layers of FCP and HCP in the BEO tundra have the highest rates of $CH_4$ oxidation. However, the highest $CH_4$ oxidation potentials were measured in the transition zone and permafrost of FCP, the
only layers with active methanogenesis. This result suggests that most active methanotrophs are found in these deeper soil layers.

Our results demonstrated $CH_4$ oxidation might not be primarily $O_2$ diffusion-limited, but rather limited by the availability of $CH_4$. The highest $CH_4$ oxidation potentials were measured below the rhizosphere, in suboxic layers where $CH_4$ has
accumulated. Given that half-saturation constants for $CH_4$ and $O_2$ used in methanotrophy models vary over 1-2 orders of magnitude (Segers, 1998; Riley et al., 2011), aerobic $CH_4$ oxidation could occur throughout much of the soil column, as advective, diffusive or plant-mediated transport processes introduce $O_2$ into the soil. Others have observed deep soil $CH_4$ oxidation activity in peatlands (Hornibrook et al., 2009), fens (Cheema et al., 2015) and wet tundra (Barbier et al., 2012), often correlated with water table depth (Sundh et al., 1994).

The water table in the center of Barrow LCPs and FCPs varies somewhat during the thaw season but remains close to the surface (<10 cm below surface for most of the thaw season) (Liljedahl et al., 2015; Liljedahl et al., 2016). Precipitation balances evapotranspiration during the thaw season, with little lateral runoff (Dingman et al., 1980), and volumetric water contents remain constant for these features (http://permafrostwatch.org). In HCPs, the water table drops up to 20 cm below
the surface following snowmelt (Liljedahl et al., 2015), and the soils have a lower volumetric water content. Due to limited drainage in the flat coastal plain, the frozen cores analyzed here are representative of field conditions for much of the thaw season. Water isotope analysis demonstrated that most water in the deep active layer comes from summer precipitation rather than seasonal ice melt (Throckmorton et al., 2016). Precipitation during September and October 2011 was above average for Barrow (http://climate.gi.alaska.edu/), suggesting a high water table during the winter freeze-up before we collected soil
cores in early 2012. Cold water from precipitation mixing in the soil column could provide methanotrophs with sufficient oxygen to grow below the rhizosphere, close to the methane source. A comparison of methanogenesis and methane oxidation potential in peat bogs demonstrated that methanotrophs survived temporary exposure to anoxic conditions, suggesting these organisms can tolerate rapid changes in the water table and redox potential (Whalen and Reeburgh, 2000). These





observations argue against the hypothesis that $CH_4$ oxidation occurs primarily at the surface layer or at the water table interface.

Temperature disparately affects $CH_4$ production and oxidation, leading to a complex response in net surface-atmosphere $CH_4$
flux. The 4.2-fold increase in $CH_4$ production in the transition zone due to a 10°C rise in temperature ($Q_{10}$) was similar to the average value of 4.26 reported in a recent meta-analysis of permafrost-affected soils (Schädel et al., 2016). These values are substantially higher than the temperature sensitivity of $CH_4$ oxidation from both freshly thawed and pre-incubated samples ($Q_{10}$ = 2.0). While both $CH_4$ production and oxidation respond positively to increased temperature, $CH_4$ production rates are predicted to increase more rapidly with higher temperature at this critical interface between the active layer and permafrost.
This difference in temperature sensitivity of $CH_4$ production and oxidation was also found in Artic lakes (Lofton et al., 2014). $CH_4$ oxidation potential from FCP showed a temperature coefficient ($Q_{10}$) between 1.7 and 2.0, which is consistent with reported values from peat (Segers, 1998). Higher $Q_{10}$ values for $CH_4$ oxidation were reported in drier mineral Arctic cryosols with low organic carbon content (Jørgensen et al., 2015; Christiansen et al., 2015). When methanotrophs are located in the upper oxic portion of the soil column they should be more susceptible to changes in air temperature than methanogens
in the lower anoxic soil layers. In the FCP studied here where both $CH_4$ production and oxidation potential are highest in the transition zone, temperature differentials should be small.

Assuming no diffusion limitation, we evaluated the net effect of $CH_4$ production and oxidation based on potential rate measurements from the transition zone of FCP that exhibited the highest $CH_4$ oxidation potential. Methane oxidation rates
are 14, 9, and 7 times of the methanogenesis rate at -2, 4, and 8°C, respectively. It is quite likely that methanogenesis will outcompete $CH_4$ oxidation under much warmer temperature if there is no decrease in soil water content. We introduced model representations to explore this possibility, as previous studies suggest $CH_4$ oxidation rate is strongly regulated by the $CH_4$ supply (Liikanen et al., 2002; Lofton et al., 2014). By assuming zero net $CH_4$ production, we estimated the temperature profile of the active biomass ratio between methanogens and methanotrophs (Fig. 7). Given the large difference in the rates
of $CH_4$ production and oxidation, $CH_4$ oxidation can still easily exceed $CH_4$ production with an active biomass ratio $B_{methanotrophs}/B_{methanogens}$ lower than 1. Studies of functional genes involved in methane production ($mcr$A) and oxidation ($pmo$A) in the active layer of four LCPs in the Western Canadian Arctic suggested substantial variation in the $pmo$A/$mcr$A abundance ratio; the range is between $8.5 \times 10^{-5}$ and $7.6 \times 10^{2}$ (Frank-Fahle et al., 2014). Future investigation using molecular methods to quantify methanogen and methanotroph populations will provide better constraints. These results suggest the
importance of parameterizing the temperature response function and biomass growth function specifically for methanogenesis and methane oxidation in model simulations to determine if the rates of methanogenesis and methane oxidation offset each other in the soil column.



The thawed soil incubations of FCP and HCP revealed substantial differences in the temporal dynamics of $CO_2$ production. An initial lag was observed from HCP samples at all temperatures, suggesting low initial microbial activity, which might be due to low moisture content, or substrate limitation. Initial active microbial biomass might also be very low given the low soil water availability. In contrast, rapid $CO_2$ production was observed from both active layer incubated under oxic

conditions and transition zone and permafrost incubated under anoxic conditions. This temporal pattern of rapid accumulation after thawing was also observed for anaerobic respiration from LCP and HCP soils (Roy Chowdhury et al., 2015; Yang et al., 2016).

Coupled iron reduction and organic carbon oxidation processes made significant contributions to total anaerobic $CO_2$

production. Acetate was most abundant and exhibited the most dynamic concentration changes among individual organic acids measured from the anaerobic microcosms, which is consistent with previous studies on LCP and HCP carbon decomposition (Yang et al., 2016). The consumption of acetate correlated with increases of $CH_4$ and $CO_2$ concentrations in previous incubation experiments suggested either acetoclastic methanogenesis or syntrophic acetate oxidation coupled to hydrogenotrophic methanogenesis, which are both consistent with isotopic analyses of $CH_4$ from the site (Throckmorton et

al., 2015; Vaughn et al., 2016). Using reaction stoichiometry for acetoclastic methanogenesis and anaerobic respiration through iron reduction (Istok et al., 2010), we estimated the amount of acetate being consumed by these parallel processes in the transition zone and permafrost (Fig. 8). In the transition zone, about half of available acetate was consumed by methanogenesis during the first 20 days of incubation, and the number dropped to about 30% during 20 to 90 days of incubation with increasing available acetate pool and increasing Fe(III) reduction. Permafrost contained a higher level of

available acetate at the beginning of the incubation, and over 60% of the available acetate was rapidly consumed by Fe(III) reduction during the first 20 days of incubation. From 20 to 90 days of incubation, iron reduction still potentially consumed over half of available acetate in the permafrost. This estimation is consistent with our initial characterization of the FCP core, where the transition zone contained the highest dissolved $CH_4$ concentrations, and permafrost was associated with significantly higher Fe(II) concentrations (Fig. S1). If hydrogen or other organic anions such as formate or propionate were

oxidized by methanogens or iron reducers, then estimated acetate production levels would decrease slightly. These simulations indicate that anaerobic iron respiration could be responsible for most of the acetate mineralized in these soil incubations.

Based on these findings, we propose the following scheme of soil carbon biogeochemistry in the FCP (Fig. 9): (1) Increasing

temperature facilitates aerobic decomposition of organic carbon in the active layer, and accelerates anaerobic carbon decomposition in the lower active layer and transition zone through fermentation, iron reduction, and methanogenesis to produce $CH_4$ and $CO_2$. (2) $CH_4$ produced in the transition zone and permafrost is oxidized close to the site of production or transported to the atmosphere. (3) Fe(III) reduction is the primary anaerobic process responsible for the depletion of acetate, the primary SOC decomposition intermediate. In this scheme, the oxic/anoxic interface could dynamically move in the soil



column with changes in the water table and porewater distribution. Although the transition zone contained much less carbon than the permafrost layer, the total carbon loss as the sum of $CO_2$ and $CH_4$ was comparable to that from permafrost. Laboratory measurements suggested that acetate accumulated in the active layer could be transported into deeper layers to support iron reduction and methanogenesis (Yang et al., 2016). This transport might occur through vertical movement of
dissolved organic compounds or mixing through cryoturbation (Drake et al., 2015).

**5 Conclusions**

Increased warming is predicted to accelerate the transition from wet LCPs to drier FCPs and HCPs in Arctic tundra, which probably function as potential atmospheric $CH_4$ sinks. This study demonstrated that $CH_4$ oxidation capacity was tightly
linked to methane availability, rather than $O_2$ availability in these soils. Thus the zone of highest $CH_4$ oxidation potential is at the suboxic area near the FCP transition zone and the upper permafrost. The measured $CH_4$ oxidation potential is an order of magnitude higher than the methanogenesis rate. With higher $CH_4$ residence time in the soil column due to limited gas diffusion in the field, $CH_4$ oxidation could easily consume $CH_4$ produced in deep permafrost soil at warming temperature. Given that iron reduction-coupled organic carbon decomposition dominates the overall anaerobic processes, $CO_2$ is likely to
remain the major form of carbon emission from degraded polygons. This finding provides critical information about the dynamics of $CH_4$ production and oxidation with increased temperature that need to be incorporated into Arctic terrestrial ecosystem models for better predictions.

**Data availability**

The dataset can be found in (Zheng et al., 2017).

**Author contributions**

DG, SW, and BG conceived and organized the research study; TRC, DG and SW collected core samples; JZ, TRC, ZY and performed experiments and acquired data; JZ, TRC and DG analysed and interpreted data; JZ and DG drafted the manuscript. All authors contributed revisions to the manuscript and have given approval to the final version of the manuscript.

**Competing interests**

The authors declare no competing interests.





**Acknowledgments**

We thank Ji-Won Moon for assistance with ion chromatography, and Bob Busey, Larry Hinzman, Kenneth Lowe and Craig Ulrich for assistance obtaining frozen core samples, as well as UMIAQ, LLC for logistical assistance. The Next-Generation Ecosystem Experiments in the Arctic (NGEE Arctic) project is supported by the Biological and Environmental Research

program in the U.S. Department of Energy (DOE) Office of Science. Oak Ridge National Laboratory is managed by UT-Battelle, LLC, for the DOE under Contract No. DE-AC05-00OR22725.

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



**Table 1.** Concentrations of Organic Acids* from FCP Transition Zone and Permafrost

| | | FCP Transition Zone | | | FCP Permafrost | | |
|---|---|---|---|---|---|---|---|
| | Incubation days | -2°C | 4°C | 8°C | -2°C | 4°C | 8°C |
| Formate | 0 | | 0.68 | | | 1.68 | |
| | 20 | 0.77 | 0.72 | 0.86 | 1.56 | 1.43 | 1.96 |
| | 90 | 0.48±0.06 | 0.52 ±0.02 | 0.44 ±0.07 | 0.99±0.2 | 1.04±0.24 | 1.14±0.05 |
| Acetate | 0 | | 1.28 | | | 10.97 | |
| | 20 | 1.44 | 1.76 | 1.89 | 10.95 | 9.78 | 12.94 |
| | 90 | 2.00±0.44 | 3.10±0.02 | 3.27±0.07 | 12.79±1.18 | 15.29±0.75 | 16.78±2.92 |
| Propionate | 0 | | 0.49 | | | 3.82 | |
| | 20 | 0.53 | 0.65 | 0.62 | 3.73 | 2.97 | 3.82 |
| | 90 | 0.51±0.12 | 0.51±0.03 | 0.39±0.06 | 8.9±0.33 | 8.8±0.18 | 10.1±0.41 |
| Butyrate | 0 | | 0.10 | | | 1.74 | |
| | 20 | 0.12 | 0.16 | 0.20 | 1.87 | 1.64 | 2.13 |
| | 90 | 0.08±0.02 | 0.15±0.01 | 0.11±0.03 | 1.92±0.11 | 2.11±0.07 | 2.17±0.13 |
| Oxalate | 0 | | 0.09 | | | 0.23 | |
| | 20 | 0.12 | 0.14 | 0.16 | 0.28 | 0.31 | 0.34 |
| | 90 | 0.10±0.02 | 0.11±0.00 | 0.08±0.01 | 0.23±0.03 | 0.24±0.03 | 0.26±0.01 |

**\*Results are presented in μmol g$^{-1}$ (on a soil dry mass basis). The average and standard deviation are shown for triplicate microcosms incubated for 90 days.**





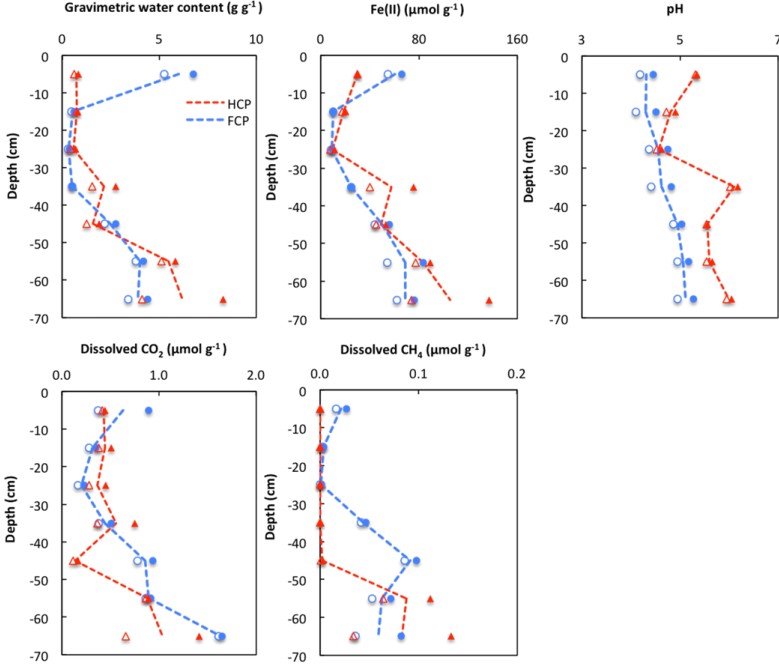

**Fig. 1. Depth profile of gravimetric water content, Fe(II) concentration, pH and soil pore water dissolved $CO_2$ and $CH_4$ concentrations from FCP (blue) and HCP (red) cores. Replicate measurements were plotted as high and low values in each soil section as filled and empty circles (FCP) and triangles (HCP). Trend lines were plotted based on the averaged value of high and low.**





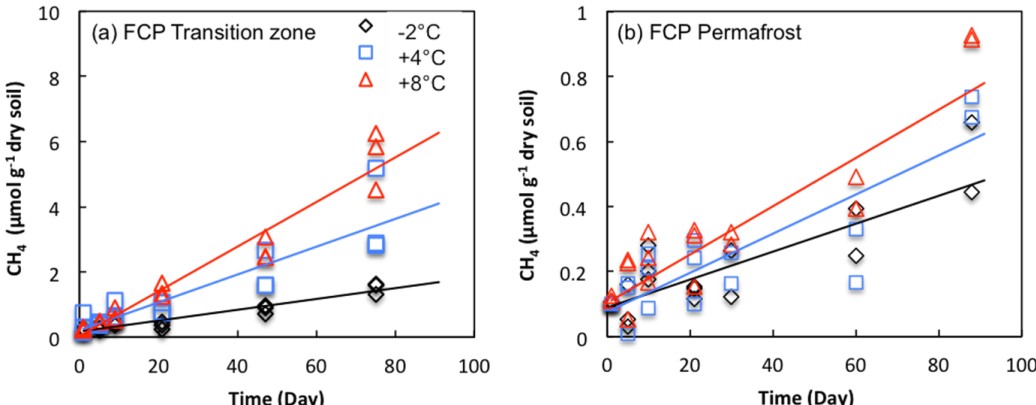

**Fig. 2. CH₄ production in anoxic soil microcosms from (a) transition zone, and (b) permafrost of the FCP at indicated temperature. Cumulative CH₄ production was standardized to dry soil mass.**

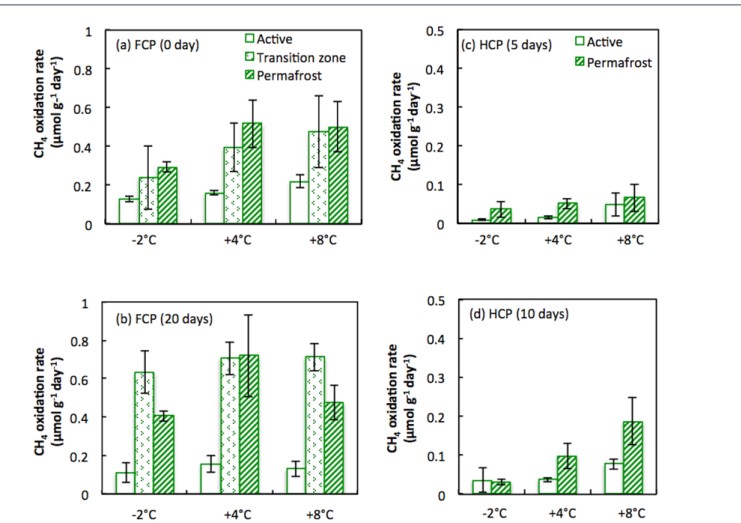

**Fig. 3. CH₄ oxidation potential measured from soils incubated at the indicated temperatures from (a) FCP after 0 days, (b) FCP after 20 days, (c) HCP after 5 days, and (d) HCP after 10 days. Error bars are ±1 sample standard deviation from the mean of three replicate incubations. Cumulative CH₄ oxidation was standardized to dry soil mass.**





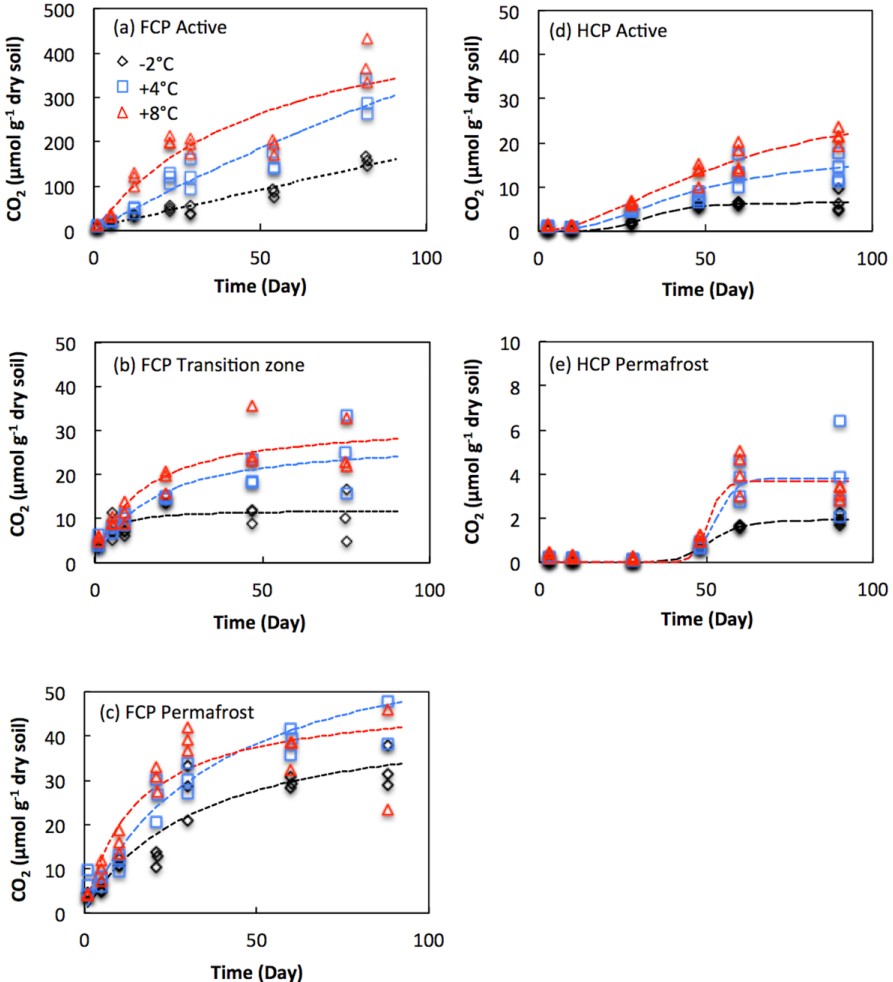

Fig. 4. Cumulative $CO_2$ production in soil microcosms from (a) active layer, (b) transition zone, and (c) permafrost of the FCP center core and (d) active and (e) permafrost layers of the HCP center core. No transition zone was identified in the HCP core.





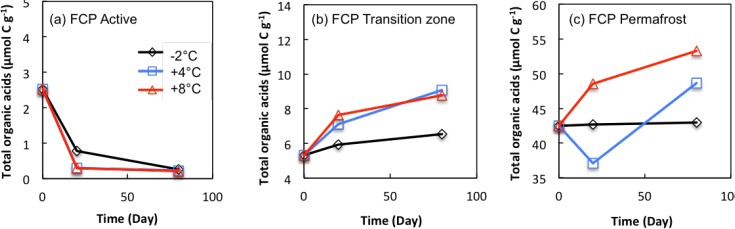

**Fig. 5. Changes in total organic acids ($T_{OA}$) in soil microcosms from (a) active layer, (b) transition zone and (c) permafrost of FCP. Total organic acids (µmol C g-1) were calculated from the concentrations of individual organic acids (Table 1).**

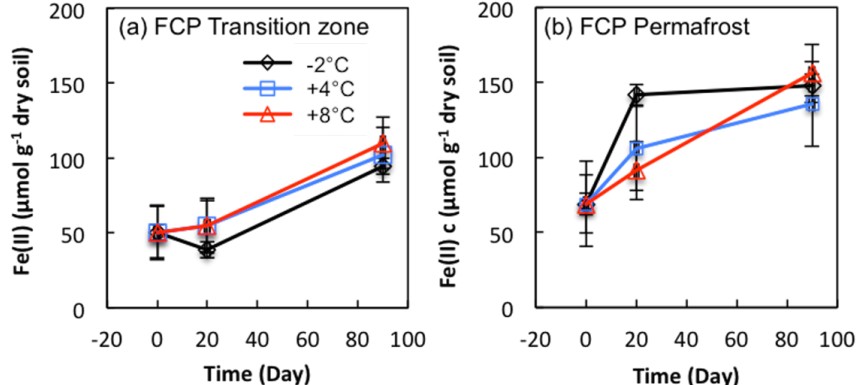

**Fig. 6. Changes in Fe(II) concentrations from (a) transition zone and (b) permafrost of FCP during anoxic incubations. Error bars are ±1 standard deviation from three replicate incubations.**




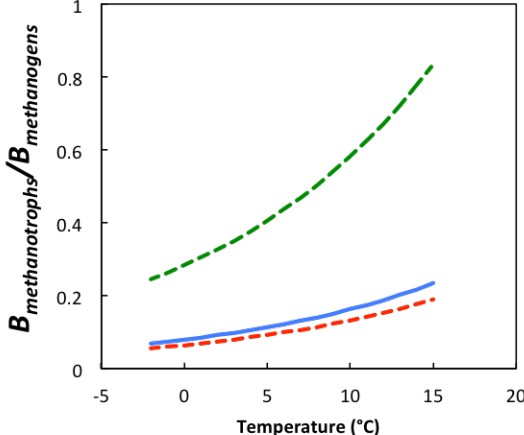

**Fig. 7.** Active biomass ratio $B_{methanotrophs}/B_{methanogens}$ needed for zero net $CH_4$ emission at different temperature. Dissolved $CH_4$ and $O_2$ concentrations in soil pore water are assumed to be 0.1mM. Half saturation rates ($K_{m,CH_4}$ and $K_{m,O_2}$) represent the baseline value (Blue), high (Green) and low(Red) range for sensitivity analysis.





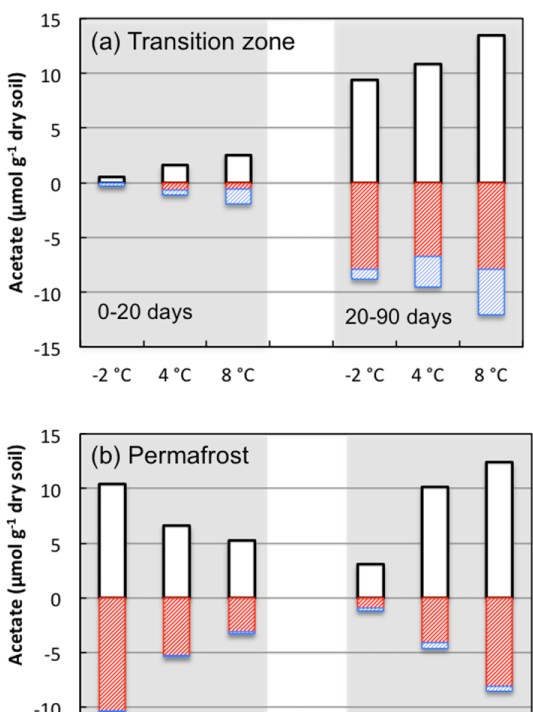

**Fig. 8. Acetate production (white bars) and consumption by iron reducing bacteria (red bars) or methanogens (blue bars) were estimated using stoichiometric calculations based on measurements of methane and Fe(II) during incubations from 0 to 20 days and from 20 to 90 days at the indicated incubation temperatures.**



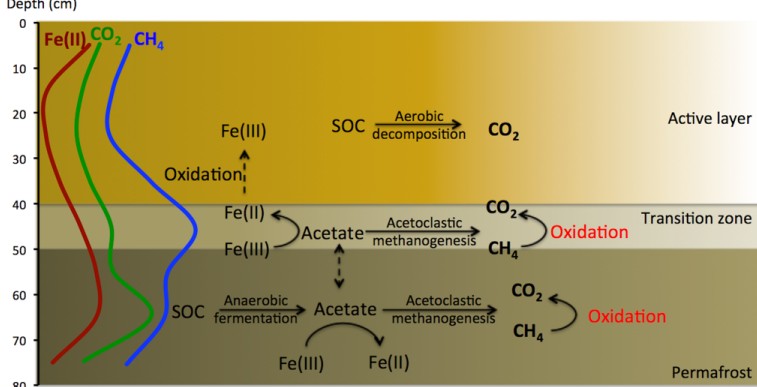

**Fig. 9. Conceptual model of aerobic and anaerobic soil organic carbon decomposition pathways and the release of $CO_2$ and $CH_4$ from a flat-centered polygon. Lines on the left marked as Fe(II), $CO_2$, and $CH_4$ represent qualitative gradients through the soil column. The median height of the water table in FCP centers is at the ground surface, with a standard deviation of ±7 cm during**
5      **the thaw season (Liljedahl et al., 2015).**