# Peer review of "Impacts of temperature and soil characteristics on methane production and oxidation in Arctic tundra"

_Biogeosciences, 2017_

## Referee Comment (RC1) · Anonymous Referee #1 · 2 Mar 2018

General Comments

This manuscript focuses on an important problem: the fate of the vast arctic carbon stores. It is unknown how much of this carbon will be released to that atmosphere as methane. However, we do know that emissions will be highly contingent on processes of methanogenesis and methane oxidation. How these processes will proceed in the Arctic is not entirely clear. This manuscript takes a sensible approach in proposing hypotheses that are based on better-known temperate systems. The hypotheses are then evaluated in the context of arctic soils.

In testing the hypotheses, the first surprising result was that methane oxidation rates

did not seem to be largest near the surface (where oxygen is most abundant). Instead, these rates were largest where methane concentrations were highest. In this way, arctic soils may differ from lower-latitude soils. This manuscript also made important comparisons between the temperature sensitivities of methane oxidation and production. Understanding these temperature sensitivities is an essential step toward understanding how methane emissions will change under a warming climate.

Overall, I think that this manuscript has the potential to be an understandable, interesting, and useful contribution to the literature. However, as it currently stands, there are some weaknesses in the methods, and the conclusions are not entirely justified. Here are a few major points:

1. It does not seem that the microcosms were controlled for soil water content. This could be a major problem: the classic understanding of methanogenesis is that there is an optimum soil moisture for methane oxidation (e.g., Zhuang et al. 2004, Global Biogeochemical Cycles, 18, GB3010). Wouldn't soil water variation confound the results? Note that soil water can vary both across samples and, through evaporation, over time in a single sample.

2. A more rigorous statistical analysis would make the results more compelling. What are the p-values of the different fits in Figure 2? Are there any patterns in the residuals?

3. Regarding hypothesis 2, the bit about production exceeding consumption is not very compelling. Doesn't production have to exceed consumption? Otherwise, wouldn't concentrations would eventually go negative? Of course, consumption can exceed production if atmospheric methane is being consumed, but I don't think the authors meant to go in that direction.

4. The text reads as if the experiment isolated the gross rates of methane production and methane consumption. However, I was not convinced that this was the case. As far as I could tell, only the net rate was evaluated. It was not clear what effect this mismatch would have on the conclusions.

5. Finally, there are numerous points (listed below) that require clarification.

Specific comments

P2, L29-30: The presence of a CH4 gradient, by itself, does not suggest that methane oxidation is being underestimated.

P3, L6: "rapid": Be more specific. Are you talking about diurnal variability, day-to-day variability, seasonal variability, something else?

Section 2.3.1: I am confused as to the number of microcosms. Is it 5x9x3 = 135? (5 soil layers x 9 replicates x 3 temperatures)? Please clarify.

Section 2.3.2: Again, I am confused as to the number of replicates. Line 3 says three replicates, line 5 says nine replicates. Also, this section is called "methane oxidation potential assay", but there are still both methanogenesis and methanotrophy going on (at least as far as I can tell). Is the argument that the effects of methanogenesis are negligible? The results would be more convincing if you explicitly make this argument.

Section 2.5: Several points need clarification. The text states that B_methanotrophs and B_methanogens were "estimated", but it does not say how they were estimated. Please clarify. The text states that Vmax,oxi and Vmeasure,pro were obtained from incubations, but does not provide details. Explain how this is done. Were all incubations at all temperatures used, or was only a subset? Also, for any given incubation, how do you separate out production and consumption (since both are presumably happening in all incubations)? What is the justification for assuming that Roxi=Rpro? Finally, the text states that initial CH4 and O2 measured concentrations were used, but don't you need a time series of these to estimate the parameters?

Section 3.2.1: Why is there apparently negligible production from the HCP permafrost soil, incubated under anoxic conditions?

P13, L23-26: These sentences are a direct description of results obtained in this study. They belong in the "Results" section.

Discussion: I am wondering if you could include a few sentences that explicitly describe how your results will effect the development of mechanistic methane models.

Technical corrections

P2, L27: "huge" is too imprecise

P2, L29: "deeper" than what?

P3, L7 and L24: Why is it a nonlinear response to temperature "fluctuations"? Isn't it a nonlinear response to temperature? (That is, I think you should omit the word "fluctuations".)

P3, L25: Respond more "rapidly" or more "strongly"?

P13, L4: "disparately" is the wrong word here.

P13, L23-24: What is meant by "temperature profile"?

---

## Referee Comment (RC2) · Anonymous Referee #2 · 29 Mar 2018

The manuscript "Impacts of temperature and soil characteristics on methane production and oxidation in Arctic polygonal tundra" of Zheng and co-authors presents results from incubation experiments of samples from two polygon centres of the arctic tundra in Alaska. The authors sectioned two cores in three layers (active layer, transition zone, permafrost) and incubated samples of these layers under either aerobic or anaerobic conditions. They measured methane ($CH_4$) production in the anaerobic layers and $CO_2$ production and $CH_4$ oxidation in all of the layers at three different temperatures (-2°C, 4°C, 8°C). Furthermore they measured low molecular weight fatty acids and ferrous iron concentrations at three time points of the incubation experiment and gradients of dissolved $CO_2$ and $CH_4$ concentrations at the field sites. From the data of the tem-

perature incubation experiments they calculated Q10 values for CH4 production and oxidation at each depth layer at the two sampling sites.

The manuscript presents potentially interesting data but the study seems not clearly focussed. The main part of the study deals with CH4 production and oxidation but one of the main novel conclusions is that iron reduction is more important for the anaerobic degradation of organic matter than methanogenesis. This would be an interesting result but the methodology and data used to support this this conclusion remain unclear. It is unclear how the authors assessed the importance of methanogenesis and iron reduction. The authors present acetate concentrations and then calculate how much of this acetate was consumed by methanogenesis and iron reduction (Fig. 8). However, it remains unclear how this was done. Acetate concentrations in the soil are a function of acetate production rates e.g. by fermentation and acetate consumption rates e.g. by methanogenesis and iron reduction. Hence concentrations give no information about production rates. Furthermore, the description of the experiments and analysis is in many parts unclear (see also specific comments). It is difficult to follow the incubation experiment and in particular the CH4 oxidation experiment. Samples were incubated at different temperatures to measure the temperature response of CH4 oxidation, but they seem to have been also pre-incubated, but at different temperatures at the different sampling sites. This is confusing and should be clarified. One of the two hypothesis rather states current knowledge than a novel research idea. Furthermore, the aim of some of the presented approaches in the manuscript remain obscure, e.g. the "calculation of net CH4 emissions" (2.5).

specific comments

P1, L23: To my knowledge, high latitude terrestrial ecosystems are a clear CH4 source, even if atmospheric CH4 may be oxidized in dry soils. Please rephrase.

P2, L14: See comment above.

P3, L12: This might be right for the oxidation of atmospheric CH4, but for wetlands,

showing substantial CH4 production, this is not the case. Generally, highest CH4 oxidation is found in wetlands at the aerobic/anaerobic interface, which is close to the water table.

P3, L32: This sentence is unclear. Why is additional research on CH4 oxidation needed to improve estimates on CH4 production? Please rephrase.

P4, L1: Please specify the carbon decomposition pathways investigated.

P4, L5: This is not a hypothesis but well established textbook knowledge.

P5 L24ff: Please clearly explain, which samples were incubated aerobically and which anaerobically. I assume the samples treated in the anaerobic chamber were also incubated anaerobically but this is not stated.

P6 L4ff: Which samples? Are this the same "microcosms" than presented in 2.3.1? and how much is ample?

P6 L9: Why are there two different incubation temperatures for FCP and HCP? I understood from the preceding sentence that the samples were incubated at the tree different temperatures -2°C, 4°C and 8°C. Please clarify.

P6, L20: Please cite the method for Fe2+ measurements.

P6, L25: Table S3.

P7, L3ff: The concept presented here is unclear. What is the aim of these calculations? Do the authors aim to calculate CH4 emissions as stated in the header? Please clarify. Furthermore, some of the assumptions are probably not met. It is unclear why the rate of CH4 oxidation should equal the rate of CH4 production? This would mean zero emission of CH4. Is this likely? And finally the authors assume a certain Km-value for CH4 and O2 and also give a very wide range of reported Km values. It should be explained why these particular Km-values were chosen. And how would a change in the Km-values affect the calculated biomass of methanogens and methanotrophs.

P7, L19ff: It would be interesting to see the water content related to soil volume. The different depth layers show substantial differences in organic carbon concentrations, which likely are also related to substantial differences in bulk densities.

P8, L1ff: Dissolved gas concentrations should be calculated based on volume soil pore water (e.g. as $\mu$M). Relating it to dry weight is misleading considering that gas cannot be dissolved in a solid.

P8, L5: If no CH4 was detected, does this indicate the oxidation of atmospheric methane in the soil? The detection limit was given as 1 ppm, which is below atmospheric concentrations.

P8, L7: Which statistical test was used to test for significance?

P8 L9ff: Better give the carbon concentrations together with the other profile data in Fig. 1. What about the carbon concentrations above 10 cm soil depth? If these are missing, a general comparison between active layer and the other samples is problematic, since generally active layer carbon concentrations are highest at the surface.

P8, L30: What means 0 and 5 days? Were they pre-incubated for 5 days with CH4? Please clearly explain in M&M.

P9, L12ff: The data on the temperature response of CH4 production and oxidation should not be presented only in the text of the manuscript but also as a graph or table as well. According to the title of the manuscript these data are the most important ones.

P9, L18ff: Please explain the meaning of the error for the Q10 values and how this was calculated.

P10, L15ff: Calculating Q10 values from rates derived from different fitting methods (linear and hyperbolic) at the respective temperatures is problematic. I suggest using only one fitting method for all of the incubation temperatures and then use these data to calculate Q10 values.

P10, L18f: Please explain how the Q10 value was estimated.

P10, L19f: This sentence should go to the discussion.

P10, L23ff: Please explain in the M&M how these fatty acids were analysed.

P10, L30: Please explain how significance tests were conducted. There seem to be no replicate analysis before day 90.

P11, L5: please explain this approach in M&M.

P11, L14: Please explain how the rates were calculated. Over the whole incubation period or only for certain incubation intervals?

P11, L15: How were Q10 values "estimated"?

P11, L28: This sentence is unclear. Why does lower active layer than permafrost CH4 concentrations indicate CH4 oxidation in the active layer? Permafrost CH4 is not released from the permafrost since it is frozen. Please clarify.

P11, L29ff: This statement is incorrect. There are numerous studies on CH4 production and CH4 oxidation in the Arctic also showing that CH4 is produced in the anoxic soil layers and oxidized in oxic soil layers. This is an obvious fact, which likely needs no further testing if there is no evidence against it. Furthermore, differences in the temperature response of CH4 production and oxidation has been shown also for Arctic environments and respective studies were also cited by the authors.

P12, L4f: This statement is not completely correct. It is current knowledge and obvious, that CH4 production depends on both CH4 and O2 supply. Therefore, indeed CH4 oxidation depends on oxygen supply but if CH4 is present. Hence, many studies on CH4 oxidation in wetlands (including those in the Arctic) demonstrate that the oxic/anoxic interface is the zone of most intense CH4 oxidation, which are not necessarily the aerobic surface soil layers, since there, as the authors correctly stated, low CH4 concentrations limit CH4 oxidation. Hence the soil water table is often more informative than the gravimetric water content for identifying the zone of maximum CH4 oxidation.

P12, L30f: The meaning of this sentence is unclear. Do the authors assume, that the main oxygen source in the saturated zone is from dissolved oxygen in rain water percolating through the soil and not from molecular transport through the gas phase through unsaturated pores? Please clarify?

P12, L34ff: Which observations? I do not see that the survival of methanotrophs under changing redox conditions argue against highest CH4 oxidation at the water table. I assume the authors mean here in situ CH4 oxidation and not potential CH4 oxidation measured in the laboratory. It has been shown repeatedly that highest CH4 oxidation is found in the soil layer where elevated CH4 concentrations overlap with oxygen. This is in soils generally close to the water table. However, if the water table fluctuates, potential CH4 oxidation rates measured in the laboratory do not need to correlate with the current water table, but likely in situ CH4 oxidation rates do. There is no way to aerobically oxidize CH4 without the presence of CH4 and oxygen.

P13, L13F: Why should this be? Please explain.

P13, L20f: What is meant by "outcompete"? Methanogens and CH4 oxidizers are not competitors. I understand that it is meant that CH4 production is expected to be higher than CH4 oxidation. But why is this likely. It has been shown that even at 8°C the potential CH4 oxidation with the current community size is 7 times higher than methanogenesis. I would rather say that it is highly unlikely that CH4 production will be higher than potential CH4 oxidation.

P13, L21-L29: This part of the discussion is unclear and in part speculative. The purpose of these calculations was not clearly stated in the description in the M&M section (see above) nor is it here. It might be interesting if the authors would have data on the microbial biomass of methanogens and CH4 oxidizers. But as it is now, it gives no substantial additional information.

P14, L1: Which incubations are referred to? The permafrost only or also the active layer?

P14, L4f: To which samples is referred to here? To the FCP samples and the HCP samples?

P15, L5f: The described pattern was obviously not observed for the HCP in this study. What could be the differences to the cited study?

P14, L9f: It is obvious that organic carbon oxidation processes contribute to anaerobic $CO_2$ production, which is the result of organic carbon oxidation. Please rephrase.

P14, L12ff: This sentence should be split into two. Furthermore, the information content is limited. It seem obvious that $CH_4$ isotopes are consistent with either acetoclastic methanogenesis or hydrogenotrophic methanogenesis since these are the mayor pathways of methanogenesis. Does this sentence mean that acetate is mainly oxidized via methanogenesis and not via iron reduction? This seems to contradict the first sentence of this paragraph.

P14, L15: These calculations should be described in the M&M section. The acetate concentrations are rising during the incubations. Hence, there is a net production over time. But how was gross acetate production calculated? This is not possible from the concentration data alone. The data presented in Fig. 8 are not comprehensible.

P14, L29ff: This last paragraph gives the current and well-established view of organic matter decomposition in wetlands. It might fit to the introduction but is not needed at the end of the discussion. The relative importance of iron reduction versus methanogenesis is an interesting issue but the data collected here does not allow a meaningful comparison of these two processes. Hence, I rather suggest omitting Fig. 9.

Fig 5: Please show in the panels which samples were incubated aerobically and which anaerobically.

Fig. 8: Acetate concentrations rather than acetate production are presented in this

Figure. Please rephrase.

Fig S1: This figure is unclear. What do the red circles mean?

---

## Author Comment (AC1) · 29 Apr 2018

*General Comments*
*This manuscript focuses on an important problem: the fate of the vast arctic carbon stores. It is unknown how much of this carbon will be released to that atmosphere as methane. However, we do know that emissions will be highly contingent on processes of methanogenesis and methane oxidation. How these processes will proceed in the Arctic is not entirely clear. This manuscript takes a sensible approach in proposing hypotheses that are based on better-known temperate systems. The hypotheses are then evaluated in the context of arctic soils.*

*In testing the hypotheses, the first surprising result was that methane oxidation rates did not seem to be largest near the surface (where oxygen is most abundant). Instead, these rates were largest where methane concentrations were highest. In this way, arctic soils may differ from lower-latitude soils. This manuscript also made important comparisons between the temperature sensitivities of methane oxidation and production. Understanding these temperature sensitivities is an essential step toward understanding how methane emissions will change under a warming climate.*

*Overall, I think that this manuscript has the potential to be an understandable, interesting, and useful contribution to the literature. However, as it currently stands, there are some weaknesses in the methods, and the conclusions are not entirely justified. Here are a few major points:*

*1. It does not seem that the microcosms were controlled for soil water content. This could be a major problem: the classic understanding of methanogenesis is that there is an optimum soil moisture for methane oxidation (e.g., Zhuang et al. 2004, Global Biogeochemical Cycles, 18, GB3010). Wouldn't soil water variation confound the results? Note that soil water can vary both across samples and, through evaporation, over time in a single sample.*

The incubated soils were kept at their original soil water content to best represent the field conditions in the thaw season in Barrow. These microcosms were created by placing soil in serum vials sealed with butyl rubber stoppers. Therefore, no changes in soil water content are expected during incubations, and we treated soil moisture as constant in individual samples during the incubation. Soil moisture was indeed significantly different among different samples, contributing to the variations in observed differences in methanogenesis and iron reduction rates. We'll add additional discussion on soil moisture in the revised manuscript. We developed a new figure illustrating the experimental design (discussed below), which should help clarify this point.

*2. A more rigorous statistical analysis would make the results more compelling. What are the p-values of the different fits in Figure 2? Are there any patterns in the residuals?*

We have fitted the data using both linear and hyperbolic models before selecting the linear model. We will add the p-values in the result section and provide a residual plot in the supplementary material.

*3. Regarding hypothesis 2, the bit about production exceeding consumption is not very compelling. Doesn't production have to exceed consumption? Otherwise, wouldn't concentrations would eventually go negative? Of course, consumption can exceed production if atmospheric methane is being consumed, but I don't think the authors meant to go in that direction.*

We consider methane consumption exceeds production when the concentration of methane in soil column is lower than the ambient level. Methane production and consumption have different temperature sensitivity, thus the net methane production in response to warming is undetermined. We will rephrase the question to clarify in the revised manuscript.

*4. The text reads as if the experiment isolated the gross rates of methane production and methane consumption. However, I was not convinced that this was the case. As far as I could tell, only the net rate was evaluated. It was not clear what effect this mismatch would have on the conclusions.*

The experiment isolated the gross rates of methane production and potential methane consumption. Gross production was measured by incubating samples in an anoxic $N_2$ headspace, while potential gross methane consumption was measured by incubating samples in ambient air with addition of 1% $CH_4$ headspace. The new figure should make the experimental design easier to understand, and we will clarify in the method sections 2.3.1 and 2.3.2.

*5. Finally, there are numerous points (listed below) that require clarification.*

*Specific comments*
*P2, L29-30: The presence of a CH4 gradient, by itself, does not suggest that methane oxidation is being underestimated.*

The discrepancy between high $CH_4$ concentrations in deep soil and near zero surface emissions suggest $CH_4$ oxidation can be an important factor determining surface $CH_4$ flux rates. We will clarify the types of gas flux estimates or models that could be affected by this discrepancy in the revised manuscript.

*P3, L6: "rapid": Be more specific. Are you talking about diurnal variability, day-to-day variability, seasonal variability, something else?*

Revision: accelerated warming.

*Section 2.3.1: I am confused as to the number of microcosms. Is it 5x9x3 = 135? (5 soil layers x 9 replicates x 3 temperatures)? Please clarify.*

Yes. We started with 5x9x3 = 135 microcosms to measure $CH_4$ and $CO_2$ production. For each soil layer x temperature combination, 3 of the 9 replicates were opened to set up $CH_4$ oxidation experiments at Day 10, and additional 3 replicates were opened at Day 20. We created a new figure for the revised manuscript to better explain the workflow (see attached).

*Section 2.3.2: Again, I am confused as to the number of replicates. Line 3 says three replicates, line 5 says nine replicates. Also, this section is called "methane oxidation potential assay", but there are still both methanogenesis and methanotrophy going on (at least as far as I can tell). Is the argument that the effects of methanogenesis are negligible? The results would be more convincing if you explicitly make this argument.*

Three replicates (about 10 g soil each) were opened to reconstruct nine methane oxidation assays (about 2 g soil each). Please see the new figure. We will clarify that methanogenesis is expected to be negligible under the fully oxic conditions of the methane oxidation potential assay.

*Section 2.5: Several points need clarification. The text states that B_methanotrophs and B_methanogens were "estimated", but it does not say how they were estimated. Please clarify. The text states that Vmax,oxi and Vmeasure,pro were obtained from incubations, but does not provide details. Explain how this is done. Were all incubations at all temperatures used, or was only a subset? Also, for any given incubation, how do you separate out production and consumption (since both are presumably happening in all incubations)? What is the justification for assuming that Roxi=Rpro? Finally, the text states that initial CH4 and O2 measured concentrations were used, but don't you need a time series of these to estimate the parameters?*

This simple simulation for Figure 7 was performed to illustrate the increasing ratio of methanotrophs to methanogens required for a zero net $CH_4$ emission scenario at increasing temperature. Therefore, we calculated the ratio of methanotroph biomass ($B\_methanotrophs$) to methanogen biomass ($B\_methanogens$) by assuming $R_{oxi}=R_{pro}$. This simulation illustrates whether the soil is going to be a $CH_4$ source or sink at $B\_methanotrophs$ to $B\_methanogens$ ratios different from these equilibrium curves. We will modify Figure 7 in the revised manuscript with clear marks of $CH_4$ source and sink: $CH_4$ sink above the plotted lines, and $CH_4$ source below the plotted lines.

$V_{max, oxi}$ and $V_{measure, pro}$ were obtained from rates measured at three temperatures in soils from the FCP transition zone, as this layer exhibited highest $CH_4$ production and consumption rates. By fitting measured rates at three different temperatures with an exponential function, we further estimated the biomass ratio in response to temperature changes. Only the initial $CH_4$ and $O_2$ concentrations are needed for assessment of methane balance in the given soil. No temporal scale is included in Figure 7. We will clarify the calculations in the revised manuscript.

*Section 3.2.1: Why is there apparently negligible production from the HCP permafrost soil, incubated under anoxic conditions?*

The measured $CH_4$ concentrations from HCP permafrost were mostly below the

detection limit of our gas chromatograph with flame ionization detector. We believe this is mostly due to the overall low microbial activity from the HCP permafrost, also measured as $CO_2$ production.

*P13, L23-26: These sentences are a direct description of results obtained in this study. They belong in the "Results" section.*

We assumed zero net $CH_4$ production to demonstrate the possible uncertainties associated with temperature increase and the sensitivity to different ratios of methane producing and consuming microbes (Figure 7). This simulation is a discussion point used to support our point that more accurate representation (and measurement) of methanotrophs and methanogens biomass is needed. We will clarify this simulation, as described above.

*Discussion: I am wondering if you could include a few sentences that explicitly describe how your results will effect the development of mechanistic methane models.*

We will add an additional paragraph discussing how to use these incubation results in mechanistic methane models.

*Technical corrections*
*P2, L27: "huge" is too imprecise*

We will provide a more quantitative assessment of the differences in the revised manuscript.

*P2, L29: "deeper" than what?*

We will add specific depths in the revised manuscript.

*P3, L7 and L24: Why is it a nonlinear response to temperature "fluctuations"? Isn't it a nonlinear response to temperature? (That is, I think you should omit the word "fluctuations".)*

We will omit "fluctuations" in the revised manuscript.

*P3, L25: Respond more "rapidly" or more "strongly"?*

We will replace "rapidly" with "strongly" in the revised manuscript.

*P13, L4: "disparately" is the wrong word here.*

We will change it to "disproportionately" in the revised manuscript.

*P13, L23-24: What is meant by "temperature profile"?*

We meant "in response to temperature change". We will rewrite that sentence in the revised manuscript.

---

## Author Comment (AC2) · 29 Apr 2018

*The manuscript "Impacts of temperature and soil characteristics on methane production and oxidation in Arctic polygonal tundra" of Zheng and co-authors presents results from incubation experiments of samples from two polygon centres of the arctic tundra in Alaska. The authors sectioned two cores in three layers (active layer, transition zone, permafrost) and incubated samples of these layers under either aerobic or anaerobic conditions. They measured methane (CH4) production in the anaerobic layers and CO2 production and CH4 oxidation in all of the layers at three different temperatures (-2_C, 4_C, 8_C). Furthermore they measured low molecular weight fatty acids and ferrous iron concentrations at three time points of the incubation experiment and gradients of dissolved CO2 and CH4 concentrations at the field sites. From the data of the temperature incubation experiments they calculated Q10 values for CH4 production and oxidation at each depth layer at the two sampling sites.*

*The manuscript presents potentially interesting data but the study seems not clearly focussed. The main part of the study deals with CH4 production and oxidation but one of the main novel conclusions is that iron reduction is more important for the anaerobic degradation of organic matter than methanogenesis. This would be an interesting result but the methodology and data used to support this this conclusion remain unclear.*

*It is unclear how the authors assessed the importance of methanogenesis and iron reduction. The authors present acetate concentrations and then calculate how much of this acetate was consumed by methanogenesis and iron reduction (Fig. 8). However, it remains unclear how this was done. Acetate concentrations in the soil are a function of acetate production rates e.g. by fermentation and acetate consumption rates e.g. by methanogenesis and iron reduction. Hence concentrations give no information about production rates. Furthermore, the description of the experiments and analysis is in many parts unclear (see also specific comments). It is difficult to follow the incubation experiment and in particular the CH4 oxidation experiment. Samples were incubated at different temperatures to measure the temperature response of CH4 oxidation, but they seem to have been also pre-incubated, but at different temperatures at the different sampling sites. This is confusing and should be clarified. One of the two hypothesis rather states current knowledge than a novel research idea. Furthermore, the aim of some of the presented approaches in the manuscript remain obscure, e.g. the "calculation of net CH4 emissions" (2.5).*

We will add more explanation on how the relative importance of iron reduction and methanogenesis is calculated based on reaction stoichiometry in the revised manuscript. We will also add a new figure to explain the workflow for anoxic incubations and methane oxidation assays (see attached figure and responses to Reviewer 1). We will also expand section 2.5 to clarify how the calculations were done in the revised

manuscript.

*specific comments*
*P1, L23: To my knowledge, high latitude terrestrial ecosystems are a clear CH4 source,*
*even if atmospheric CH4 may be oxidized in dry soils. Please rephrase.*

We will rephrase in the revised manuscript to clarify the uncertainty over which soils will
function as a net source or sink in high latitude ecosystems.

*P2, L14: See comment above.*

*P3, L12: This might be right for the oxidation of atmospheric CH4, but for wetlands,*
*showing substantial CH4 production, this is not the case. Generally, highest CH4*
*oxidation is found in wetlands at the aerobic/anaerobic interface, which is close to the*
*water table.*

We will rephrase in the revised manuscript with a better distinction between submerged
(wetland) soils and unsaturated soils.

*P3, L32: This sentence is unclear. Why is additional research on CH4 oxidation needed*
*to improve estimates on CH4 production? Please rephrase.*

We will rephrase in the revised manuscript.

*P4, L1: Please specify the carbon decomposition pathways investigated.*

We will specify the pathways we measured, including fermentation, iron reduction and
methanogenesis in the revised manuscript. This will include the added detail on reaction
stoichiometry, discussed above.

*P4, L5: This is not a hypothesis but well established textbook knowledge.*

We will specify the hypothesis in the context of flat centered polygons and high centered
polygons, which have relatively dry organic layers and wet permafrost layers, in the
revised manuscript.

*P5 L24ff: Please clearly explain, which samples were incubated aerobically and which*
*anaerobically. I assume the samples treated in the anaerobic chamber were also*
*incubated anaerobically but this is not stated.*

Organic soils were incubated under oxic conditions, while soils from transition zone and
permafrost were incubated under anoxic conditions. The new figure should clarify this
important point, and we will elaborate in the revised manuscript.

*P6 L4ff: Which samples? Are this the same "microcosms" than presented in 2.3.1?*
*and how much is ample?*

A subset of microcosms setup in section 2.3.1 were opened to set up methane oxidation
assays. We will clarify in section 2.3.2 and the additional figure to demonstrate how the
anaerobic incubations and methane oxidation assays were constructed.

*P6 L9: Why are there two different incubation temperatures for FCP and HCP? I understood from the preceding sentence that the samples were incubated at the tree different temperatures -2_C, 4_C and 8_C. Please clarify.*

The samples were incubated at three different temperatures. We put a subset of samples on the shaker to remove gas-liquid phase transfer limitation. We will clarify in the revised manuscript.

*P6, L20: Please cite the method for Fe2+ measurements.*

We will add the citation to this commercial assay in the revised manuscript.

*P6, L25: Table S3.*

Will revise.

*P7, L3ff: The concept presented here is unclear. What is the aim of these calculations? Do the authors aim to calculate CH4 emissions as stated in the header? Please clarify. Furthermore, some of the assumptions are probably not met. It is unclear why the rate of CH4 oxidation should equal the rate of CH4 production? This would mean zero emission of CH4. Is this likely? And finally the authors assume a certain Km-value for CH4 and O2 and also give a very wide range of reported Km values. It should be explained why these particular Km-values were chosen. And how would a change in the Km-values affect the calculated biomass of methanogens and methanotrophs.*

We will rewrite section 2.5 to clarify. (Please see also comments in response to Reviewer 1). The aim of the simulation is to demonstrate the wide range of uncertainties in net methane production and impact of methanotroph to methanogen biomass ratios in response to temperature increase. We assumed zero net $CH_4$ production to help us understanding whether the soil is predicted to be a $CH_4$ source or sink. We will change Figure 7 in the revised manuscript with clear marks of $CH_4$ source and sink: $CH_4$ sink above the plotted lines, and $CH_4$ source below the plotted lines.
We intentionally included a wide range of $K_m$ values used in models for this sensitivity analysis as we do not have enough information to preferably select certain $K_m$ values. We will clarify in the revised manuscript and add additional lines of discussion.

*P7, L19ff: It would be interesting to see the water content related to soil volume. The different depth layers show substantial differences in organic carbon concentrations, which likely are also related to substantial differences in bulk densities.*

We will add a plot of soil bulk density as an additional panel in Figure 1.

*P8, L1ff: Dissolved gas concentrations should be calculated based on volume soil pore water (e.g. as _M). Relating it to dry weight is misleading considering that gas cannot be dissolved in a solid.*

We will also include molar calculations of dissolved gas relative to volume soil pore water. Normalizing gas concentrations to soil mass facilitates stoichiometric comparisons with organic acids, iron, and gases produced in microscosms, although it is

not physically relevant.

*P8, L5: If no CH4 was detected, does this indicate the oxidation of atmospheric methane in the soil? The detection limit was given as 1 ppm, which is below atmospheric concentrations.*

No. This data can only be interpreted as no $CH_4$ produced was measured. We infer this observation is due to the low level of total microbial activity measured as $CO_2$ production.

*P8, L7: Which statistical test was used to test for significance?*

We used a paired t-test. Will clarify in the revised manuscript.

*P8 L9ff: Better give the carbon concentrations together with the other profile data in Fig. 1. What about the carbon concentrations above 10 cm soil depth? If these are missing, a general comparison between active layer and the other samples is problematic, since generally active layer carbon concentrations are highest at the surface.*

We will add an SOC subplot in Figure 1.

*P8, L30: What means 0 and 5 days? Were they pre-incubated for 5 days with CH4? Please clearly explain in M&M.*

Pre-incubated without $CH_4$. The added figure illustrating the experimental workflow should clarify this point.

*P9, L12ff: The data on the temperature response of CH4 production and oxidation should not be presented only in the text of the manuscript but also as a graph or table as well. According to the title of the manuscript these data are the most important ones.*

We will add a figure on the temperature response of $CH_4$ production and oxidation in the revised manuscript.

*P9, L18ff: Please explain the meaning of the error for the Q10 values and how this was calculated.*

We will clarify in the revised manuscript.

*P10, L15ff: Calculating Q10 values from rates derived from different fitting methods (linear and hyperbolic) at the respective temperatures is problematic. I suggest using only one fitting method for all of the incubation temperatures and then use these data to calculate Q10 values.*
*P10, L18f: Please explain how the Q10 value was estimated.*

We used linear fitting to estimate the initial production rate of $CO_2$ for Q10 calculation. We will clarify in the revised manuscript.

*P10, L19f: This sentence should go to the discussion.*

Will move this sentence.

*P10, L23ff: Please explain in the M&M how these fatty acids were analysed.*

We will add more explanation of organic acid analysis in the revised manuscript.

*P10, L30: Please explain how significance tests were conducted. There seem to be no replicate analysis before day 90.*

We used paired t-test with additional technical replicates.

We will add the explanation in the revised manuscript.

*P11, L14: Please explain how the rates were calculated. Over the whole incubation period or only for certain incubation intervals?*

Iron reduction rates were estimated by the changes in Fe(II) concentration. We will clarify in the revised manuscript.

The Q10 values of iron reduction were estimated using the ratio of iron reduction rate measured at 8 degree C and -2 degree C. We will clarify in the revised manuscript.

*P11, L28: This sentence is unclear. Why does lower active layer than permafrost CH4 concentrations indicate CH4 oxidation in the active layer? Permafrost CH4 is not released from the permafrost since it is frozen. Please clarify.*

We clarify in the revised manuscript.

*P11, L29ff: This statement is incorrect. There are numerous studies on CH4 production and CH4 oxidation in the Arctic also showing that CH4 is produced in the anoxic soil layers and oxidized in oxic soil layers. This is an obvious fact, which likely needs no further testing if there is no evidence against it. Furthermore, differences in the temperature response of CH4 production and oxidation has been shown also for Arctic environments and respective studies were also cited by the authors.*

We will rephrase the questions to be more specific to polygonal tundra with fine scale microtopographic features.

*P12, L4f: This statement is not completely correct. It is current knowledge and obvious, that CH4 production depends on both CH4 and O2 supply. Therefore, indeed CH4 oxidation depends on oxygen supply but if CH4 is present. Hence, many studies on CH4 oxidation in wetlands (including those in the Arctic) demonstrate that the oxic/anoxic interface is the zone of most intense CH4 oxidation, which are not necessarily the aerobic surface soil layers, since there, as the authors correctly stated, low CH4 concentrations limit CH4 oxidation. Hence the soil water table is often more informative than the gravimetric water content for identifying the zone of maximum CH4 oxidation.*

We will clarify in revision that both $CH_4$ and $O_2$ diffusion can limit aerobic methane oxidation. We appreciate the reviewer's perspective on the importance of the oxic/anoxic interface as the hotspot for aerobic $CH_4$ oxidation in wetlands. A cited review by Segers (1998) provides a valuable overview of potential methane oxidation rates as a function of distance to oxic/anoxic interface (p. 39). Average rates are highest near the water table as expected, but maximum values are on the anoxic side of the interface. However, this distance factor explains only a small part of the variance in observed in the distribution of methane oxidation potential. Therefore, other factors must influence methane oxidation potential as well.

We are still surprised that the maximum methane oxidation potential in flat-centered polygon soils was observed in the transition (40-50 cm) and permafrost (50-70 cm) layers –far below the near-surface water table and overlapping with areas of anaerobic methanogenesis and iron reduction. One could interpret this as a result of a fluctuating water table (as the reviewer suggests, below). However, there is no evidence for recent fluctuations in the near-surface water table at this flat center polygon, as discussed on page 12. Alternatively, we could hypothesize that the oxic/anoxic interface comprises a large part of this soil column rather than the narrow horizontal line usually drawn near the water table in conceptual diagrams. Such a broad suboxic zone would be consistent with the dissolved Fe(II) and $CH_4$ profiles show in in Figure 1. Proximity to $CH_4$ sources would be more important than proximity to the water table in this model. Future studies will be required to understand the complex $O_2$ transport mechanisms in this cold, saturated FCP soil. We will clarify this discussion in the revised manuscript.

*P12, L30f: The meaning of this sentence is unclear. Do the authors assume, that the main oxygen source in the saturated zone is from dissolved oxygen in rain water percolating through the soil and not from molecular transport through the gas phase through unsaturated pores? Please clarify?*

Based on the high water table of flat-centered polygons and the substantial precipitation preceding our sampling campaign, we do not expect much gas transport through unsaturated pores in this soil. We will clarify this point in revision.

*P12, L34ff: Which observations? I do not see that the survival of methanotrophs under changing redox conditions argue against highest CH4 oxidation at the water table. I assume the authors mean here in situ CH4 oxidation and not potential CH4 oxidation measured in the laboratory. It has been shown repeatedly that highest CH4 oxidation is found in the soil layer where elevated CH4 concentrations overlap with oxygen. This is in soils generally close to the water table. However, if the water table fluctuates, potential CH4 oxidation rates measured in the laboratory do not need to correlate with the current water table, but likely in situ CH4 oxidation rates do. There is no way to aerobically oxidize CH4 without the presence of CH4 and oxygen.*

See response above.

*P13, L13F: Why should this be? Please explain.*

Sharp temperature gradients along soil depth.

*P13, L20f: What is meant by "outcompete"? Methanogens and CH4 oxidizers are*

*not competitors. I understand that it is meant that CH4 production is expected to be higher than CH4 oxidation. But why is this likely. It has been shown that even at 8_C the potential CH4 oxidation with the current community size is 7 times higher than methanogenesis. I would rather say that it is highly unlikely that CH4 production will be higher than potential CH4 oxidation.*

We will replace "outcompete" in the revised manuscript to better describe the kinetics of these two processes. Our point in the simulation shown in Figure 7 is to illustrate the disparate effects of temperature on methanogenesis and methane oxidation activity and address model sensitivity to assumptions of half saturation rates. We will use an example to clarify interpretation of this figure.

*P13, L21-L29: This part of the discussion is unclear and in part speculative. The purpose of these calculations was not clearly stated in the description in the M&M section (see above) nor is it here. It might be interesting if the authors would have data on the microbial biomass of methanogens and CH4 oxidizers. But as it is now, it gives no substantial additional information.*

See above.

*P14, L1: Which incubations are referred to? The permafrost only or also the active layer?*

This refers to all incubations, including permafrost and active layer.

*P14, L4f: To which samples is referred to here? To the FCP samples and the HCP samples?*

FCP samples. We will clarify in the revised manuscript.

*P15, L5f: The described pattern was obviously not observed for the HCP in this study. What could be the differences to the cited study?*

We did not see evidence of cryoturbation in the HCP core used in this study. The organic layer of HCP contained much lower level of organic acids comparing to the FCP organic layer, so overall the substrate level is low.

*P14, L9f: It is obvious that organic carbon oxidation processes contribute to anaerobic CO2 production, which is the result of organic carbon oxidation. Please rephrase.*

We will rephrase in the revised manuscript to distinguish decomposition from mineralization processes.

*P14, L12ff: This sentence should be split into two. Furthermore, the information content is limited. It seem obvious that CH4 isotopes are consistent with either acetoclastic methanogenesis or hydrogenotrophic methanogenesis since these are the mayor pathways of methanogenesis. Does this sentence mean that acetate is mainly oxidized via methanogenesis and not via iron reduction? This seems to contradict the first sentence of this paragraph.*

We will revise this sentence to more clearly justify the use of acetoclastic methanogenesis reaction stoichiometry.

*P14, L15: These calculations should be described in the M&M section. The acetate concentrations are rising during the incubations. Hence, there is a net production over time. But how was gross acetate production calculated? This is not possible from the concentration data alone. The data presented in Fig. 8 are not comprehensible.*

We will clarify the calculation and provide an example. The net production of acetate over time was measured. The consumption of acetate was calculated based on the stoichiometry of iron reduction and methanogenesis utilizing acetate as electron donor. Thus we estimated the overall gross production of acetate.

*P14, L29ff: This last paragraph gives the current and well-established view of organic matter decomposition in wetlands. It might fit to the introduction but is not needed at the end of the discussion. The relative importance of iron reduction versus methanogenesis is an interesting issue but the data collected here does not allow a meaningful comparison of these two processes. Hence, I rather suggest omitting Fig. 9.*

We believe the conceptual figure 9 will help readers to integrate the numerous processes discussed in this paper. Therefore, we prefer to keep it.

*Fig 5: Please show in the panels which samples were incubated aerobically and which anaerobically.*

We will clarify in the revised manuscript and the new figure.

*Fig. 8: Acetate concentrations rather than acetate production are presented in this Figure. Please rephrase.*

We will revise the Figure caption in the revised manuscript.

*Fig S1: This figure is unclear. What do the red circles mean?*

The circles show the combined soil sections used for incubations. We will clarify in the revised manuscript.

---

## Author Response (AR1)

We appreciate comments from both reviewers, and have used these to extensively revise our manuscript. This document includes responses to the reviewers' comments. Finally, we have attached a comparison of the revised manuscript with the originally submitted version, at the end of these responses. These changes were widespread and substantially improved the manuscript's readability. Note that figures have been renumbered in the revised version.

*Anonymous Referee #1*

*General Comments*
*This manuscript focuses on an important problem: the fate of the vast arctic carbon stores. It is unknown how much of this carbon will be released to that atmosphere as methane. However, we do know that emissions will be highly contingent on processes of methanogenesis and methane oxidation. How these processes will proceed in the Arctic is not entirely clear. This manuscript takes a sensible approach in proposing hypotheses that are based on better-known temperate systems. The hypotheses are then evaluated in the context of arctic soils.*

*In testing the hypotheses, the first surprising result was that methane oxidation rates did not seem to be largest near the surface (where oxygen is most abundant). Instead, these rates were largest where methane concentrations were highest. In this way, arctic soils may differ from lower-latitude soils. This manuscript also made important comparisons between the temperature sensitivities of methane oxidation and production. Understanding these temperature sensitivities is an essential step toward understanding how methane emissions will change under a warming climate.*

*Overall, I think that this manuscript has the potential to be an understandable, interesting, and useful contribution to the literature. However, as it currently stands, there are some weaknesses in the methods, and the conclusions are not entirely justified. Here are a few major points:*

*1. It does not seem that the microcosms were controlled for soil water content. This could be a major problem: the classic understanding of methanogenesis is that there is an optimum soil moisture for methane oxidation (e.g., Zhuang et al. 2004, Global Biogeochemical Cycles,*

*18, GB3010). Wouldn't soil water variation confound the results?Note that soil water can vary both across samples and, through evaporation, over time in a single sample.*

The incubated soils were kept at their original soil water content to best represent the field conditions in the thaw season in Barrow. These microcosms were created by placing soil in serum vials sealed with butyl rubber stoppers. Therefore, no changes in soil water content are expected during incubations, and we treated soil moisture as constant in individual samples during the incubation. Soil moisture was indeed significantly different among different samples, contributing to the variations in observed differences in methanogenesis and iron reduction rates. We developed a new Figure 1 illustrating the experimental design (discussed below), which should help clarify this point.

*2. A more rigorous statistical analysis would make the results more compelling. What are the p-values of the different fits in Figure 2? Are there any patterns in the residuals?*

We have fitted the data using both linear and hyperbolic models before selecting the linear model. The revised figure (now Fig. 3) includes a 95% confidence interval for the linear regression model. There was no apparent trend in the residuals from this regression.

*3. Regarding hypothesis 2, the bit about production exceeding consumption is not very compelling. Doesn't production have to exceed consumption? Otherwise, wouldn't concentrations would eventually go negative? Of course, consumption can exceed production if atmospheric methane is being consumed, but I don't think the authors meant to go in that direction.*

We consider methane consumption exceeds production when the concentration of methane in soil column is lower than the ambient level. Methane production and consumption have different temperature sensitivity, thus the net methane production in response to warming is undetermined. We clarified the second hypothesis to focus on this differential temperature sensitivity in the last paragraph of Section 1.

*4. The text reads as if the experiment isolated the gross rates of methane production and methane consumption. However, I was not convinced that this was the case. As far as I could tell, only the net rate was evaluated. It was not clear what effect this mismatch would have on the conclusions.*

The experiment isolated the gross rates of methane production and potential methane consumption. Gross production was measured by incubating samples in an anoxic $N_2$ headspace, while potential gross methane consumption was measured by incubating samples in ambient air with addition of 1% $CH_4$ headspace. We added a new Figure 1 that clarifies the experimental design, and we revised Method sections 2.2 and 2.3.

*5. Finally, there are numerous points (listed below) that require clarification.*

*Specific comments*
*P2, L29-30: The presence of a CH4 gradient, by itself, does not suggest that methane*
*oxidation is being underestimated.*

The discrepancy between high $CH_4$ concentrations in deep soil and near zero surface emissions suggest $CH_4$ oxidation can be an important factor determining surface $CH_4$ flux rates. We revised the final lines of the second introductory paragraph to better explain this idea.

*P3, L6: "rapid": Be more specific. Are you talking about diurnal variability, day-to-day*
*variability, seasonal variability, something else?*

Revision: accelerated warming.

*Section 2.3.1: I am confused as to the number of microcosms. Is it 5x9x3 = 135? (5*
*soil layers x 9 replicates x 3 temperatures)? Please clarify.*

Yes. We started with 5x9x3 = 135 microcosms to measure $CH_4$ and $CO_2$ production. For each soil layer
20  x temperature combination, 3 of the 9 replicates were opened to set up $CH_4$ oxidation experiments at Day 10, and additional 3 replicates were opened at Day 20. We created a new Figure 1 for the revised manuscript to better explain the workflow.

25  *Section 2.3.2: Again, I am confused as to the number of replicates. Line 3 says three*
*replicates, line 5 says nine replicates. Also, this section is called "methane oxidation*
*potential assay", but there are still both methanogenesis and methanotrophy going on*
*(at least as far as I can tell). Is the argument that the effects of methanogenesis are*
*negligible? The results would be more convincing if you explicitly make this argument.*

Three replicates (about 10 g soil each) were opened to reconstruct nine methane oxidation assays (about 2 g soil each). We added the new Figure 1 and explained in Section 2.4 that methanogenesis is expected to be negligible under the fully oxic conditions of the methane oxidation potential assay.

35  *Section 2.5: Several points need clarification. The text states that B_methanotrophs*
*and B_methanogens were "estimated", but it does not say how they were estimated.*
*Please clarify. The text states that Vmax,oxi and Vmeasure,pro were obtained from*
*incubations, but does not provide details. Explain how this is done. Were all incubations*
*at all temperatures used, or was only a subset? Also, for any given incubation, how do*
40  *you separate out production and consumption (since both are presumably happening*

*in all incubations)? What is the justification for assuming that Roxi=Rpro? Finally, the text states that initial CH4 and O2 measured concentrations were used, but don't you need a time series of these to estimate the parameters?*

5 This simple simulation for Figure 7 (now Figure 8) was performed to illustrate the increasing ratio of methanotrophs to methanogens required for a zero net $CH_4$ emission scenario at increasing temperature. Therefore, we calculated the ratio of methanotroph biomass ($B\_methanotrophs$) to methanogen biomass ($B\_methanogens$) at an equilibrium state where $R_{oxi}=R_{pro}$. This simulation illustrates whether the soil is going to be a $CH_4$ source or sink at $B\_methanotrophs$ to $B\_methanogens$ ratios different from these

10 equilibrium curves. We modified Figure 8 with text illustrating the $CH_4$ sink conditions above the plotted lines, and $CH_4$ source conditions below the plotted lines.

$V_{max,\ oxi}$ and $V_{measure,\ pro}$ were obtained from rates measured at three temperatures in soils from the FCP transition zone, as this layer exhibited highest $CH_4$ production and consumption rates. By fitting

15 measured rates at three different temperatures with an exponential function, we further estimated the biomass ratio in response to temperature changes. Only the initial $CH_4$ and $O_2$ concentrations are needed for assessment of methane balance in the given soil. No temporal scale is included in this figure. We will clarified calculations for these curves in a revised Methods section 2.6.

*Section 3.2.1: Why is there apparently negligible production from the HCP permafrost soil, incubated under anoxic conditions?*

The measured $CH_4$ concentrations from HCP permafrost were mostly below the detection limit of our

25 gas chromatograph with flame ionization detector. We believe this is mostly due to the overall low microbial activity from the HCP permafrost, also measured as $CO_2$ production.

*P13, L23-26: These sentences are a direct description of results obtained in this study. They belong in the "Results" section.*

We assumed zero net $CH_4$ production to demonstrate the possible uncertainties associated with temperature increase and the sensitivity to different ratios of methane producing and consuming microbes (Figure 8). This simulation is a discussion point used to support our point that more accurate representation (and measurement) of methanotrophs and methanogens biomass is needed. We clarified

35 this simulation, as described above.

*Discussion: I am wondering if you could include a few sentences that explicitly describe how your results will effect the development of mechanistic methane models.*

40 We added a sentence to the Discussion outlining our strategy to use results from these experiments to

structure and parameterize a mechanistic model of anaerobic organic matter decomposition with greenhouse gas production.

*Technical corrections*
*P2, L27: "huge" is too imprecise*

We replaced "huge" with "hundred-fold."

*P2, L29: "deeper" than what?*

We described these depths as deeper than 20 cm per the Vaughn et al. reference.

*P3, L7 and L24: Why is it a nonlinear response to temperature "fluctuations"? Isn't it a nonlinear response to temperature? (That is, I think you should omit the word "fluctuations".)*

We omitted "fluctuations" in the revised manuscript.

*P3, L25: Respond more "rapidly" or more "strongly"?*

We replace "rapidly" with "strongly."

*P13, L4: "disparately" is the wrong word here.*

We revised this paragraph to remove the sentence in question.

*P13, L23-24: What is meant by "temperature profile"?*

This section has been re-written for clarity.

*Anonymous Referee #2*

*The manuscript "Impacts of temperature and soil characteristics on methane production and oxidation in Arctic polygonal tundra" of Zheng and co-authors presents results from incubation experiments of samples from two polygon centres of the arctic tundra in Alaska. The authors sectioned two cores in three layers (active layer, transition zone, permafrost) and incubated samples of these layers under either aerobic or anaerobic conditions. They measured methane ($CH_4$) production in the anaerobic layers and $CO_2$ production and $CH_4$ oxidation in all of the layers at three different temperatures (-2_C, 4_C, 8_C). Furthermore they measured low molecular weight fatty acids and ferrous iron concentrations at three*

*time points of the incubation experiment and gradients of dissolved CO2 and CH4 concentrations at the field sites. From the data of the temperature incubation experiments they calculated Q10 values for CH4 production and oxidation at each depth layer at the two sampling sites.*

5   *The manuscript presents potentially interesting data but the study seems not clearly focussed. The main part of the study deals with CH4 production and oxidation but one of the main novel conclusions is that iron reduction is more important for the anaerobic degradation of organic matter than methanogenesis. This would be an interesting result but the methodology and data used to support this this conclusion remain unclear.*

*It is unclear how the authors assessed the importance of methanogenesis and iron reduction. The authors present acetate concentrations and then calculate how much of this acetate was consumed by methanogenesis and iron reduction (Fig. 8).*
*However, it remains unclear how this was done. Acetate concentrations in the soil are*
15  *a function of acetate production rates e.g. by fermentation and acetate consumption rates e.g. by methanogenesis and iron reduction. Hence concentrations give no information about production rates. Furthermore, the description of the experiments and analysis is in many parts unclear (see also specific comments). It is difficult to follow the incubation experiment and in particular the CH4 oxidation experiment. Samples were incubated at different temperatures to measure the temperature response of*
20  *CH4 oxidation, but they seem to have been also pre-incubated, but at different temperatures at the different sampling sites. This is confusing and should be clarified. One of the two hypothesis rather states current knowledge than a novel research idea. Furthermore, the aim of some of the presented approaches in the manuscript remain obscure, e.g.*
*the "calculation of net CH4 emissions" (2.5).*

We added details for the stochiometric calculations used for Figure 8 (now Figure 9) and present an example in the Discussion on page 13. A new Figure 1 illustrates the workflow for anoxic incubations and methane oxidation assays (see also responses to Reviewer 1).

*specific comments*
*P1, L23: To my knowledge, high latitude terrestrial ecosystems are a clear CH4 source,*
*even if atmospheric CH4 may be oxidized in dry soils. Please rephrase.*

We rephrased this sentence in the revised manuscript to clarify the uncertainty over which soils will function as a net CH4 source or sink in future high latitude ecosystems, affected by warming and associated hydrological changes.

40  *P2, L14: See comment above.*

We removed this sentence, which was redundant to the abstract.

*P3, L12: This might be right for the oxidation of atmospheric CH4, but for wetlands,*
*showing substantial CH4 production, this is not the case. Generally, highest CH4 oxidation is found in*
*wetlands at the aerobic/anaerobic interface, which is close to the*
*water table.*

We rephrased this sentence to better distinguish between unsaturated soils and submerged (wetland)
soils.

*P3, L32: This sentence is unclear. Why is additional research on CH4 oxidation needed to improve*
*estimates on CH4 production? Please rephrase.*

We removed this sentence.

*P4, L1: Please specify the carbon decomposition pathways investigated.*

We rewrote this sentence to better introduce the manuscript. This manuscript does not attempt to
elucidate all carbon decomposition pathways. Rather it focuses on the terminal fermentation,
methanogenesis and anaerobic respiration pathways that produce CO2 and CH4.

*P4, L5: This is not a hypothesis but well established textbook knowledge.*

We specify the hypothesis in the context of flat centered polygons and high centered polygons, which
have relatively dry organic layers and wet permafrost layers, in the revised manuscript.

*P5 L24ff: Please clearly explain, which samples were incubated aerobically and which*
*anaerobically. I assume the samples treated in the anaerobic chamber were also incubated anaerobically*
*but this is not stated.*

The new Figure 1 illustrates how organic soils were incubated under oxic conditions, while soils from
transition zone and permafrost were incubated under anoxic conditions.

*P6 L4ff: Which samples? Are this the same "microcosms" than presented in 2.3.1?*
*and how much is ample?*

The new Figure 1 and additional experimental details added to Methods section 2.3 describe how the
anaerobic incubations and methane oxidation assays were constructed.

*P6 L9: Why are there two different incubation temperatures for FCP and HCP? I understood from the preceding sentence that the samples were incubated at the tree different temperatures -2_C, 4_C and 8_C. Please clarify.*

The samples were incubated at three different temperatures. A subset of samples were shaken to minimize potential gas-liquid phase transfer limitations. This has been clarified in Methods section 2.4.

We added a description of the Hach method 8146 assay for Fe(II) to Methods section 2.1.

Corrected to Table S1.

*P7, L3ff: The concept presented here is unclear. What is the aim of these calculations? Do the authors aim to calculate CH4 emissions as stated in the header? Please clarify. Furthermore, some of the assumptions are probably not met. It is unclear why the rate of CH4 oxidation should equal the rate of CH4 production? This would mean zero emission of CH4. Is this likely? And finally the authors assume a certain Km-value for CH4 and O2 and also give a very wide range of reported Km values. It should be explained why these particular Km-values were chosen. And how would a change in the Km-values affect the calculated biomass of methanogens and methanotrophs.*

We rewrote this section 2.5 (now 2.6) to clarify. (Please see also comments in response to Reviewer 1). The aim of the simulation is to demonstrate the wide range of uncertainties in net methane production and impact of methanotroph to methanogen biomass ratios in response to temperature increase. We assumed zero net $CH_4$ production to help us understanding whether the soil is predicted to be a $CH_4$ source or sink. We changed Figure 7 (now Figure 8) in the revised manuscript with clear marks of $CH_4$ source and sink: $CH_4$ sink above the plotted lines, and $CH_4$ source below the plotted lines. We intentionally included a wide range of $K_m$ values used in models for this sensitivity analysis as we do not have enough information to preferably select certain $K_m$ values.

*P7, L19ff: It would be interesting to see the water content related to soil volume. The different depth layers show substantial differences in organic carbon concentrations, which likely are also related to substantial differences in bulk densities.*

A revised Figure 2 now illustrates both gravimetric water content and bulk density. We will add a plot of soil bulk density as an additional panel in Figure 1. As the reviewer suggests, high organic carbon content

usually leads to a low bulk density for Arctic soils.

*P8, L1ff: Dissolved gas concentrations should be calculated based on volume soil pore water (e.g. as _M). Relating it to dry weight is misleading considering that gas cannot be dissolved in a solid.*

The manuscript now reports dissolved gas concentrations in micromolar units. Normalizing gas concentrations to soil mass facilitates stoichiometric comparisons with organic acids, iron, and gases produced in microcosms, although we agree it is not physically relevant.

*P8, L5: If no CH4 was detected, does this indicate the oxidation of atmospheric methane in the soil? The detection limit was given as 1 ppm, which is below atmospheric concentrations.*

No. The limit of detection reflects the GC system's signal to noise properties. It does not reflect the efficiency of quantifying $CH_4$ dissolved in the soil pore water or headspace. This data can only be interpreted as no $CH_4$ produced was measured. We infer this observation is due to the low level of total microbial activity measured as $CO_2$ production.

*P8, L7: Which statistical test was used to test for significance?*

We clarified in section 3.1 that deeper soil layers in the polygon contained 5-7 fold higher concentrations of Fe(II) than surface layers.

*P8 L9ff: Better give the carbon concentrations together with the other profile data in Fig. 1. What about the carbon concentrations above 10 cm soil depth? If these are missing, a general comparison between active layer and the other samples is problematic, since generally active layer carbon concentrations are highest at the surface.*

Soil carbon concentrations were measured in the combined, homogenized core segments, reported in Table S1. Above 10 cm, the HCP and FCP cores contained mostly plant material, litter and ice (Figure 2 and Section 3.2). Since these segments have little soil, they were not considered here.

*P8, L30: What means 0 and 5 days? Were they pre-incubated for 5 days with CH4? Please clearly explain in M&M.*

The new Figure 1 and revised Methods section 2.4 should clarify this experimental workflow.

*P9, L12ff: The data on the temperature response of CH4 production and oxidation should not be presented only in the text of the manuscript but also as a graph or table*

*as well. According to the title of the manuscript these data are the most important ones.*

The new Figure 5 illustrates the temperature sensitivity of CH4 production and oxidation.

*P9, L18ff: Please explain the meaning of the error for the Q10 values and how this was calculated.*

A revised Methods section 2.5 describes Q10 value calculations.

*P10, L15ff: Calculating Q10 values from rates derived from different fitting methods (linear and hyperbolic) at the respective temperatures is problematic. I suggest using only one fitting method for all of the incubation temperatures and then use these data to calculate Q10 values.*
*P10, L18f: Please explain how the Q10 value was estimated.*

We used linear fitting to estimate the initial production rate of $CO_2$ for Q10 calculation. A revised Methods section 2.5 describes Q10 value calculations.

*P10, L19f: This sentence should go to the discussion.*

Revised.

*P10, L23ff: Please explain in the M&M how these fatty acids were analysed.*

A short description of the organic acid analyses was added to Methods section 2.3.

*P10, L30: Please explain how significance tests were conducted. There seem to be no replicate analysis before day 90.*

We used paired t-test with additional technical replicates.

*P11, L5: please explain this approach in M&M.*

Explained in Methods section 2.3.

*P11, L14: Please explain how the rates were calculated. Over the whole incubation period or only for certain incubation intervals?*

Iron reduction rates were estimated by the changes in Fe(II) concentration –now explained in Methods section 2.3.

*P11, L15: How were Q10 values "estimated"?*

The Q10 values of iron reduction were estimated using the ratio of iron reduction rate measured at 8 degree C and -2 degree C. Please see section 2.5.

*P11, L28: This sentence is unclear. Why does lower active layer than permafrost CH4 concentrations indicate CH4 oxidation in the active layer? Permafrost CH4 is not released from the permafrost since it is frozen. Please clarify.*

We clarify in the revised manuscript.

*P11, L29ff: This statement is incorrect. There are numerous studies on CH4 production and CH4 oxidation in the Arctic also showing that CH4 is produced in the anoxic soil layers and oxidized in oxic soil layers. This is an obvious fact, which likely needs no further testing if there is no evidence against it. Furthermore, differences in the temperature response of CH4 production and oxidation has been shown also for Arctic environments and respective studies were also cited by the authors.*

We rephrased the questions to be more specific to polygonal tundra with fine scale microtopographic features. Our data show a surprising overlap between maximum methanogenesis rates and methane oxidation potential in the transitional layer.

*P12, L4f: This statement is not completely correct. It is current knowledge and obvious, that CH4 production depends on both CH4 and O2 supply. Therefore, indeed CH4 oxidation depends on oxygen supply but if CH4 is present. Hence, many studies on CH4 oxidation in wetlands (including those in the Arctic) demonstrate that the oxic/anoxic interface is the zone of most intense CH4 oxidation, which are not necessarily the aerobic surface soil layers, since there, as the authors correctly stated, low CH4 concentrations limit CH4 oxidation. Hence the soil water table is often more in- formative than the gravimetric water content for identifying the zone of maximum CH4 oxidation.*

We clarify in revision that both $CH_4$ and $O_2$ diffusion can limit aerobic methane oxidation. We appreciate the reviewer's perspective on the importance of the oxic/anoxic interface as the hotspot for aerobic $CH_4$ oxidation in wetlands. A cited review by Segers (1998) provides a valuable overview of potential methane oxidation rates as a function of distance to oxic/anoxic interface (p. 39). Average rates are highest near the water table as expected, but maximum values are on the anoxic side of the interface. However, this

distance factor explains only a small part of the variance observed in the distribution of methane oxidation potential. Therefore, other factors must influence methane oxidation potential as well.

We are still surprised that the maximum methane oxidation potential in flat-centered polygon soils was observed in the transition (40-50 cm) and permafrost (50-70 cm) layers –far below the near-surface water table and overlapping with areas of anaerobic methanogenesis and iron reduction. One could interpret this as a result of a fluctuating water table (as the reviewer suggests, below). However, there is no evidence for recent fluctuations in the near-surface water table at this flat center polygon, as discussed on page 12. Alternatively, we could hypothesize that the oxic/anoxic interface comprises a large part of this soil column rather than the narrow horizontal line usually drawn near the water table in conceptual diagrams. Such a broad suboxic zone would be consistent with the dissolved Fe(II) and $CH_4$ profiles show in in Figure 1. Proximity to $CH_4$ sources would be more important than proximity to the water table in this model. Future studies will be required to understand the complex $O_2$ transport mechanisms in this cold, saturated FCP soil.

*P12, L30f: The meaning of this sentence is unclear. Do the authors assume, that the main oxygen source in the saturated zone is from dissolved oxygen in rain water percolating through the soil and not from molecular transport through the gas phase through unsaturated pores? Please clarify?*

Based on the high water table of flat-centered polygons and the substantial precipitation preceding our sampling campaign, we do not expect much gas transport through unsaturated pores in this soil. We clarified this point in the Discussion.

*P12, L34ff: Which observations? I do not see that the survival of methanotrophs under changing redox conditions argue against highest CH4 oxidation at the water table. I assume the authors mean here in situ CH4 oxidation and not potential CH4 oxidation measured in the laboratory. It has been shown repeatedly that highest CH4 oxidation is found in the soil layer where elevated CH4 concentrations overlap with oxygen. This is in soils generally close to the water table. However, if the water table fluctuates, potential CH4 oxidation rates measured in the laboratory do not need to correlate with the current water table, but likely in situ CH4 oxidation rates do. There is no way to aerobically oxidize CH4 without the presence of CH4 and oxygen.*

See response above.

*P13, L13F: Why should this be? Please explain.*

Sharp temperature gradients along unsaturated soil depth. This sentence has been removed to better focus the paragraph.

*P13, L20f: What is meant by "outcompete"? Methanogens and CH4 oxidizers are not competitors. I understand that it is meant that CH4 production is expected to be higher than CH4 oxidation. But why is this likely. It has been shown that even at 8_C the potential CH4 oxidation with the current community size is 7 times higher than methanogenesis. I would rather say that it is highly unlikely that CH4 production will be higher than potential CH4 oxidation.*

We replaced "outcompete" with "outpace" to better represent the kinetics of these two processes. Our point in the simulation shown in Figure 7 (now Figure 8) is to illustrate the disparate effects of temperature on methanogenesis and methane oxidation activity and address model sensitivity to assumptions of half saturation rates. We believe the simulations illustrated in Figure 8 are important to illustrate the potential impact of temperature on changes in microbial population dynamics required to maintain a CH4 equilibrium.

*P13, L21-L29: This part of the discussion is unclear and in part speculative. The purpose of these calculations was not clearly stated in the description in the M&M section (see above) nor is it here. It might be interesting if the authors would have data on the microbial biomass of methanogens and CH4 oxidizers. But as it is now, it gives no substantial additional information.*

See above.

*P14, L1: Which incubations are referred to? The permafrost only or also the active layer?*

This refers to all incubations, including permafrost and active layer. Clarified in the Discussion.

*P14, L4f: To which samples is referred to here? To the FCP samples and the HCP samples?*

FCP samples. Clarified in the Discussion.

*P15, L5f: The described pattern was obviously not observed for the HCP in this study. What could be the differences to the cited study?*

We did not see evidence of cryoturbation in the HCP core used in this study. The organic layer of HCP contained much lower level of organic acids comparing to the FCP organic layer, so overall the substrate level is low.

*P14, L9f: It is obvious that organic carbon oxidation processes contribute to anaerobic*
*CO2 production, which is the result of organic carbon oxidation. Please rephrase.*

We rephrased this section of the discussion to highlight the substantial contribution of anaerobic respiration from iron reduction to CO2 production, compared to methanogenesis and fermentation that also produce CO2.

10   *P14, L12ff: This sentence should be split into two. Furthermore, the information content*
*is limited. It seem obvious that CH4 isotopes are consistent with either acetoclastic*
*methanogenesis or hydrogenotrophic methanogenesis since these are the mayor pathways of*
*methanogenesis. Does this sentence mean that acetate is mainly oxidized via methanogenesis and not*
*via iron reduction? This seems to contradict the first sentence of this paragraph.*

This long sentence was intended to explain our assumption of acetoclastic methanogensis for our simulation. However, it introduced needless confusion through the remote possibility of complex isotope fractionation caused by syntrophic organic acid oxidation coupled to hydrogenotrophic methanogenesis. We deleted this complex sentence for clarity.

*P14, L15: These calculations should be described in the M&M section. The acetate*
*concentrations are rising during the incubations. Hence, there is a net production over*
*time. But how was gross acetate production calculated? This is not possible from the*
*concentration data alone. The data presented in Fig. 8 are not comprehensible.*

We clarified these calculations in Section 2.6 and added an example to help interpret Figure 8 (now Figure 9). The net production of acetate over time was measured. The consumption of acetate was calculated based on the stoichiometry of iron reduction and methanogenesis utilizing acetate as electron donor. Thus, we estimated the overall gross production of acetate.

*P14, L29ff: This last paragraph gives the current and well-established view of organic*
*matter decomposition in wetlands. It might fit to the introduction but is not needed at*
*the end of the discussion. The relative importance of iron reduction versus methanogenesis is an*
*interesting issue but the data collected here does not allow a meaningful comparison of these two*
35   *processes. Hence, I rather suggest omitting Fig. 9.*

We believe the conceptual Figure 9 (now Figure 10) will help readers to integrate the numerous processes discussed in this paper. Therefore, we prefer to keep it.

40   *Fig 5: Please show in the panels which samples were incubated aerobically and which*

*anaerobically.*

The new Figure 1 clarifies this experimental design.

*Fig. 8: Acetate concentrations rather than acetate production are presented in this Figure. Please rephrase.*

We revised the Figure 9 legend to described changes in acetate concentrations associated with production and consumption processes.

*Fig S1: This figure is unclear. What do the red circles mean?*

The circles show the combined soil sections used for incubations. The Figure S1 legend has been revised to explain this point.

*Changes to Manuscript*

[revised manuscript text omitted]

Fig. 1

---

## Author Response (AR2)

We appreciate both referee's reviewing our revised manuscript and are glad that our changes have adequately addressed Referee #1's comments. And we have used the thoughtful comments from new Referee #3 to clarify methodology in the second revision of this paper. This document includes responses to Referee #3's comments. Finally, we have attached a comparison of the revised manuscript with changes tracked from the originally submitted version.

*Anonymous Referee #1*

*No comments submitted for this version of the manuscript.*

*Anonymous Referee #3*

*This manuscript reports effects of temperature and soil layers on CH4 production and oxidation rates in the arctic tundra with a soil incubation experiment. I have major comments regarding the incubation methods, which lead to the results and conclusions:*

*1) The authors introduced that low-centered polygons (LCPs) are wetter than flat-centered polygons (FCPs) and high-centered polygons (HCPs) (P4). Then why not taking samples from wetter LCPs?*

We previously measured and observed high rates of methanogenesis and $CO_2$ production in wet and saturated LCP samples (Roy Chowdhury et al., 2015). Field measurements of $CH_4$ fluxes from surface soils demonstrated higher $CH_4$ emissions from saturated LCPs than drier FCPs or HCPs, but steeper gradients in porewater methane concentrations in FCP/HCP features (e.g. Vaughn et al., 2016). Therefore, we focused our aerobic $CH_4$ oxidation experiments on FCP samples that were expected to be more significant sources of $CH_4$ oxidation activity. This rationale has been clarified in the second paragraph of the Introduction.

*2) It is not clear if all soil samples are incubated anaerobically (closed lids)? If not, soil moisture would change over the 90 days of incubation, and thus changing the CH4 oxidation rates. If yes, the aerobic condition would be totally different from that in the field. Also, would the CH4 concentration in the air space be saturated and inhibit further production of CH4?*

We modified methods section 2.3 to clarify that "All vials were sealed with butyl rubber septa and crimp sealed to prevent evaporation and gas exchange." Soil moisture did not change in these sealed vials. The headspace gases for the incubation experiments were selected to match the soil's redox properties and are described in section 2.3 and illustrated in Figure 1.

Due to the low solubility of $CH_4$ in water, it is unusual for $CH_4$ accumulation to inhibit methanogenesis in a liquid-gas system. In our incubations, $CH_4$ accumulated with a linear rate profile (Figure 3) showing no indication of product inhibition or even substrate limitation. For comparison, biogas from anaerobic digestors frequently reaches 60-70% $CH_4$ concentration –orders of magnitude higher than our incubations.

*3) P5. L15. Soil samples were homogenized. Then the soil structure was damaged, compared with that in the field. How does soil texture affect the CH4 production and consumption?*

The Roy Chowdhury et al. (2015) reference was added to section 2.3 to explain the homogenization process, along with a short description of the homogenization process and its effects on the soil structure. Our method using an oscillating power tool inside a glove box disrupts large, frozen soil clumps and removes gravel or litter, allowing us to place representative soil samples inside the serum vials. This method does not affect the soil microaggregate structure or expose the samples to oxidation, drying or significant warming that could disrupt anaerobic microbial processes. We recognize that incubation experiments do not reproduce the gradients and transport processes of soil pedons in the field (see below). The small soil sample volume used in our microcosms is unlikely to present diffusion constraints, and we assume equilibrium with headspace in our measurements. Our experimental set-up, therefore, could focus on the specific question of understanding the temperature sensitivity of the distinct processes of interest, i.e. methane production and potential oxidation.

*4) How does the microbial community differ (in diversity and density) during the incubation from those in the field? This shift may change the Q10 function if the incubation results would be used to predict the field condition.*

We agree that increased temperatures are likely to change microbial community composition, both in incubations and in the field (see for example Hoj et al., 2007, ISME J. 2:37 and Conrad et al. 2009, Environ. Microbiol. 11:1844). However, $Q_{10}$ relationships are strictly empirical representations of a processes' temperature sensitivity. Microbial community changes, differences in gene expression and enzyme production, and fundamental Arrhenius enzyme kinetics are all grouped together in this term, along with any changes in sorption or transport rates. Current process-enabled models do not distinguish among these factors. To reduce the influence of microbial biomass changes on $Q_{10}$ calculations, we use a ratio of maximum activity rates, which are usually highest at the beginning of activity or immediately after a lag period (see Fig. 6 and our extended discussion in Roy Chowdhury et al., 2015).

Future manipulation experiments in the field and novel model structures and parameterization may provide a framework to quantify changes in microbial community composition from laboratory and field warming experiments and begin to differentiate mechanisms underlying $Q_{10}$ relationships. We added a short comment on this issue on pp. 12-13 of the revised manuscript.

*Minor comments:*
*Title: remove "polygon", which causes confusion to readers.*

Done.

*P2. L14. It is unclear here if the "source strength" means the carbon producing CO2 or CH4.*

This sentence has been revised for clarity: "… a current estimate of net $CH_4$ exchange from tundra to the atmosphere ranging widely from 8 to 29 Tg C $yr^{-1}$."

*P3. L7. "In the classical model of CH4 oxidation profiles" –need references.*

We added a citation to the Le Mer and Roger (2001) review whose conceptual model of methane cycling in wetlands has been highly reproduced. A reference to the metatranscriptomic study by Kim and Liesack (2015) was also added to this paragraph.

*P4. L13. Describe the diameter of the "SIPRE auger".*

We added the 3-inch inner diameter dimension of the core liner to this description. Additional details about this auger are published in Roy Chowdhury et al. (2015).

*P12. L26. The diffusion should be discussed. This also relates to my previous comment that the soil structure and text determine the diffusion of gas fluxes (CO2 and CH4). Thus, the change in diffusivity may lead to changes in gas effluxes and the response to temperature.*

This discussion section uses observed $Q_{10}$ values to develop an example simulation of competing methane oxidation and methanogenesis rates in the thin FCP transitional layer. It illustrates the impact of different process temperature sensitivities on potential $CH_4$ fluxes and microbial biomass ratios in a warming environment.

A variety of transport processes in the soil column will certainly affect surface gas fluxes. However, our discovery that maximum rates of methanogenesis and methane oxidation occur in the same thin transitional layer of FCP soil enables this simple simulation that neglects diffusion. We added a corollary to this section reminding readers that future simulations will need to include a full range of gas transport processes in order to integrate soil horizons and model net surface gas fluxes.

[revised manuscript text omitted]